# A dual-selective thermal emitter with enhanced subambient radiative cooling performance

Xueke Wu [1], Jinlei Li [2], Fei Xie [3], Xun-En Wu[4], Siming Zhao[1], Qinyuan Jiang [1], Shiliang Zhang[1], Baoshun Wang[1], Yunrui Li[1], Di Gao[1], Run Li[1], Fei Wang[1], Ya Huang[1], Yanlong Zhao [1], Yingying Zhang [4], Wei Li [3], Jia Zhu [2] & Rufan Zhang [1] ✉

Radiative cooling is a zero-energy technology that enables subambient cooling by emitting heat into outer space (~3 K) through the atmospheric transparent windows. However, existing designs typically focus only on the main atmospheric transparent window (8–13 μm) and ignore another window (16–25 μm), under-exploiting their cooling potential. Here, we show a dual-selective radiative cooling design based on a scalable thermal emitter, which exhibits selective emission in both atmospheric transparent windows and reflection in the remaining mid-infrared and solar wavebands. As a result, the dual-selective thermal emitter exhibits an ultrahigh subambient cooling capacity (~9 °C) under strong sunlight, surpassing existing typical thermal emitters (≥3 °C cooler) and commercial counterparts (as building materials). Furthermore, the dual-selective sample also exhibits high weather resistance and color compatibility, indicating a high practicality. This work provides a scalable and practical radiative cooling design for sustainable thermal management.

The last decade has witnessed a tremendous development in the daytime radiative cooling (RC), which achieves subambient cooling under sunlight without any energy consumption and greenhouse gas emissions[1–4]. It is therefore a promising and sustainable alternative to traditional energy-intensive cooling systems. The subambient cooling capability of a daytime RC material stems from its ability to emit heat (as a thermal emitter) into cold outer space through the atmospheric transparent windows, while rejecting the solar irradiation (with high solar reflectance) at the same time[5–7]. Various efficient thermal emitters have been developed as daytime RC materials for building and personal thermal management[1,5,8–20]. According to the spectral response in the mid-infrared (MIR) waveband, these thermal emitters can be mainly categorized into non-selective thermal emitters[9–12,21] and selective thermal emitters[5,20,22,23]. The non-selective thermal emitters exhibit high absorption/emission over the entire MIR wavebands (Fig. 1a (top), b), while the selective thermal emitters exhibit a high absorption/emission only in the 8–13 μm atmospheric window (defined as mono-selective RC materials) and a high reflection in the remaining MIR wavebands (Fig. 1a (middle), b and Supplementary Fig. 1a).

Compared with non-selective thermal emitters, selective thermal emitters have previously been demonstrated to have superior cooling

[1]Beijing Key Laboratory of Green Chemical Reaction Engineering and Technology, Department of Chemical Engineering, Tsinghua University, Beijing, China. [2]National Laboratory of Solid State Microstructures, College of Engineering and Applied Sciences, Jiangsu Key Laboratory of Artificial Functional Materials, Collaborative Innovation Center of Advanced Microstructures, Nanjing University, Nanjing, China. [3]GPL Photonics Laboratory, State Key Laboratory of Luminescence and Applications, Changchun Institute of Optics, Fine Mechanics and Physics, Chinese Academy of Sciences, Changchun, Jilin, PR China. [4]Key Laboratory of Organic Optoelectronics and Molecular Engineering of the Ministry of Education, Department of Chemistry, Tsinghua University, Beijing, PR China. ✉e-mail: zhangrufan@tsinghua.edu.cn

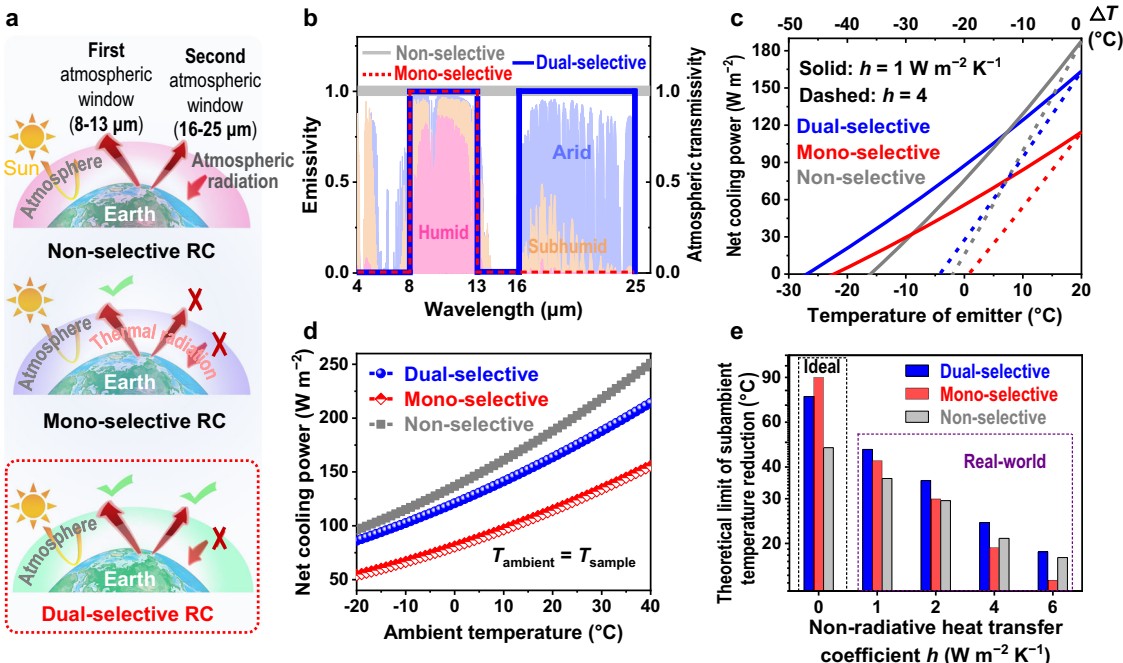

**Fig. 1 | Conceptual explanation and model calculations of dual-selective radiative cooler. a** Schematic of the radiative heat transfer process of a non-selective (top), a mono-selective (middle), and a dual-selective (bottom) radiative cooler. **b** Corresponding spectral features in the mid-infrared region for the three different types of RC models. The backgrounds are the atmospheric transmissivity of arid, sub-humid, and humid climates, respectively. The dual-selective RC model is featured with high absorption/emission occurring only within the two atmospheric windows and high reflectance outside the two windows, including solar and non-window MIR wavebands. Atmosphere transmittance data were generated using the ATRAN modeling software in the 4−25 μm waveband. **c** Theoretical cooling power of these three different radiative coolers as a function of the emitter temperature at a fixed $T_{ambient}$ of 20 °C (for the detailed calculation equations, see Supplementary text 1). The theoretical limit of subambient cooling is reached when the cooling power reaches zero (the intersection of the curve with the *x*-axis). The calculation is based on the atmospheric transmittance in an arid environment (relative humidity RH < 2%, blue in **c**) and the non-radiative heat transfer coefficient of $h = 1$ (solid lines) and 4 (dashed lines) W m$^{-2}$ K$^{-1}$. **d** Comparison of the theoretical cooling power between dual-selective, mono-selective, and non-selective radiative coolers at different $T_{ambient}$ (−20° to 40 °C). The calculation is based on the same atmospheric transmittance with **b** and the same temperature between the thermal emitter $T_{sample}$ and the $T_{ambient}$. **e** Theoretical limit of subambient cooling of the three types of radiative coolers at different $h$ (0−6 W m$^{-2}$ K$^{-1}$). The calculation is based on a net cooling power $P_{net}$ of 0 W m$^{-2}$ and $T_{ambient}$ = 20 °C. Source data are provided as a Source Data file.

performance due to the exclusion of downward atmospheric parasitic heat[5,20,24]. However, existing mono-selective RC materials only exhibit superior cooling performance in ideal or near-ideal outdoor scenarios, where non-radiative heat can be neglected (Fig. 1c−e and Supplementary Fig. 2). In most real scenarios (with a non-radiative heat transfer coefficient $h \geq 2$ W m$^{-2}$ K$^{-1}$), the cooling performance of the mono-selective emitters usually sharply degrades and even becomes worse than that of the non-selective emitters (Fig. 1e and Supplementary Fig. 3). Obviously, the cooling effect of mono-selective emitters is more easily to be compromised by non-radiative heat because of its lower cooling power than the non-selective emitters (50−100 W m$^{-2}$ lower, Supplementary Fig. 4), which limits their practical applications (e.g., low color compatibility)[25−27].

Actually, in addition to the widely known atmospheric transparent window in the 8−13 μm waveband, there is also an atmosphere transparent window in the 16−25 μm waveband (defined as the second atmospheric window), especially in arid climates (Fig. 1a and Supplementary Fig. 5)[2,3,24,28−31]. As is known, there are increasingly urgent cooling demands in hot arid areas or seasons throughout the world[32,33]. However, the cooling effect of the second atmospheric window is often overlooked in most previous studies because its atmospheric transparency decreases with increasing humidity due to the strong heat absorption by water in this waveband[5,34]. To date, there is still a lack of explicit theoretical research and material design for the second atmospheric window, which limits the cooling potential of RC materials, especially in arid climates.

Here, we show a dual-selective RC design based on a scalable polymer-metal thermal emitter that achieves selective heat emission in

both the first and second atmospheric transparent windows, enabling enhanced radiative cooling performance over existing typical thermal emitters. Crucially, the resulting dual-selective thermal emitter exhibits an emittance of 83.2% in the first atmospheric window (8−13 μm), an emittance of 67.5% in the second window (16−25 μm), and a high reflectance in the remaining MIR and solar wavebands (a solar reflectance of 95.4% in 0.3−2.5 μm), making it an ideal dual-selective RC material. As a result, the dual-selective sample exhibits ultrahigh sub-ambient cooling performance (8.7 ± 0.8 °C) under strong sunlight (> 700 W m$^{-2}$), which is notably higher than that of typical non-selective and mono-selective thermal emitters (5.0 ± 0.8 °C and 5.8 ± 1.0 °C, respectively). Furthermore, the dual-selective thermal emitter also exhibits a high net cooling power (151.8 ± 13.1 W m$^{-2}$) under strong sunlight (peak > 850 W m$^{-2}$), which is much higher than the mono-selective sample (87.9 ± 9.4 W m$^{-2}$). Moreover, the dual-selective thermal emitter also shows an high durability (enduring for 300 h of intense UV testing, 5 months of outdoor exposure, and 12 h of continuous water treatment). When used as a building roof, it shows notably better cooling performance than the conventional commercial counterparts. Besides, even when the dual-selective thermal emitters are covered with different colors, it still exhibits high subambient cooling performance, which is favorable for their widespread application in building roofs.

## Results

### Dual-selective radiative cooling model
To evaluate the cooling potential of the second atmospheric window, we demonstrated a dual-selective RC model featured with a selective

high absorption/emission in the two atmospheric windows and a high reflection in the remaining non-window MIR and solar wavebands (Fig. 1a, b and Supplementary Fig. 1b). Atmospheric transparency data in an arid climate (obtained from the ATRAN modeling software with a water vapor column of 1.0 mm and zero zenith angle) were used in the calculation to maximize the cooling effect of the second atmospheric window[2,35]. The theoretical models of non-selective and mono-selective emitters described above were employed for comparison (Fig. 1b). Theoretical calculations show that the dual-selective emitter has a notably higher net cooling power (120.9–214.2 W m$^{-2}$ at ambient temperature $T_{ambient}$ of 0–40 °C) than the mono-selective emitter (80.7–155.7 W m$^{-2}$), which is close to the non-selective model (136.7–250.9 W m$^{-2}$) (Fig. 1d, Supplementary text 1 and Supplementary Fig. 4). In an ideal environment ($h = 0$ W m$^{-2}$ K$^{-1}$), the dual-selective RC model has a subambient cooling performance close to that of the mono-selective model (~80 °C) and much better than the non-selective model (~50 °C) (Fig. 1c, e, Supplementary text 2 and Supplementary Fig. 2). However, in real scenarios ($h \geq 1$ W m$^{-2}$ K$^{-1}$)[36,37], the subambient cooling performance of the dual-selective RC model is notably better than that of the non-selective and mono-selective RC models. For example, at $h = 1$–6 W m$^{-2}$ K$^{-1}$, the dual-selective RC model was 1.0–10.9 °C cooler than the non-selective model, and 4.2–5.5 °C cooler than the mono-selective model (Fig. 1e, Supplementary text 3 and Supplementary Fig. 3, and Supplementary Table 1). The simulations in a higher humidity environment show that even in a high-humidity environment the dual-selective model still maintains a subambient cooling performance close to the mono-selective model and notably better than the non-selective model (Supplementary text 4 and Supplementary Figs. 5–7). Therefore, the dual-selective design is able to fully exploit the cooling potential of RC materials and thus is expected to improve the multifunctional compatibility of RC technology in practical applications (e.g., color compatibility). However, despite the existence of various dual-selective designs[38–43], no such desired dual-selective materials have yet been experimentally developed before. It remains a challenge to design a dual-selective thermal emitter through a simple method that has precise spectral control of the two atmospheric windows, non-window MIR, and solar wavebands.

## Material design and characterization

The design of the dual-selective RC emitters is based on the molecular vibration theory (molecular design in the MIR waveband of 4–25 µm) and the Mie scattering theory (nano-micron hierarchical design in the solar waveband of 0.3–2.5 µm). Specifically, the presence of strong molecular vibrations is a prerequisite for achieving strong absorption/ emission of organic materials in the corresponding wavebands[44–46]. The appropriate molecular vibrations (molecular bonds/functional groups) with the desired spectral responses are readily accessible by using the currently established "Molecular Bond vs. Wavelength" databases (e.g., NIST Standard Database)[47,48]. For an ideal dual-selective thermal emitter, the vibrational wavebands of its molecular bonds/functional groups should be limited within the two atmospheric window wavebands (8–13 µm and 16–25 µm, respectively). Based on a careful screening and analysis of the molecular bonds of common polymers and the corresponding vibrational wavebands, the combination of two widely used polymers, polyformaldehyde (POM) and polytetrafluoroethylene (PTFE), which mainly contain C-O-C and C-F bonds, respectively, is expected to provide the desired dual-selective characteristics. This is because the former has strong molecular vibrations mainly in the first atmospheric window (8–13 µm) due to the strong absorption of C-O-C bond, while the latter has strong molecular vibrations mainly in the second atmospheric window (16–25 µm) due to the strong absorption of C-F bond, and neither has strong molecular vibrations in the non-window MIR wavebands. Their molecular vibration properties were further confirmed by their Attenuated Total Reflection Fourier Transform Infrared spectra (ATR-FTIR, Fig. 2a and

Supplementary Table 2)[49]. In addition to the suitable molecular vibrations, both polymers were chosen because of their appropriate physic-chemical properties (e.g., non-toxicity, high thermal stability, and high mechanical strength) and commercial availability, which provide the basis for the practical application and scaling-up of the dual-selective RC materials. Finally, given the MIR transmission properties of polymers (transparent to heat) in the non-vibrational wavebands, a heat-reflective metal substrate (e.g., aluminum (Al) foil) is also required to achieve a high reflection in the non-window MIR wavebands.

To achieve a high cooling performance, the dual-selective thermal emitters should also have a high solar reflectance, which mainly depends on their micro/nano sizes[18]. Commercial POM-PTFE composites are typically melt-processed and lack micro/nanostructures, which inevitably results in low solar reflectance[50,51]. Therefore, a rational structural design is required for the POM and PTFE. As nano- or micron-sized POM fibers are readily achievable by electrospinning and PTFE micro/nanoparticles are commercially available (Supplementary Fig. 8)[52–54], the combination of POM fibers and PTFE particles to achieve dual-selective properties is simple and feasible. Based on this, we calculated the scattering efficiency of POM fibers and PTFE particles as a function of particle/fiber diameter across the solar spectrum (0.3–2.5 µm) using the Mie scattering theory (Fig. 2b and Supplementary text 5)[19,20,55]. As shown in Fig. 2b, a strong scattering (high reflectance) in the solar waveband can be achieved when the fiber or particle diameter distribution is close to the solar waveband (200–3000 nm), providing a clear guide for hierarchically structural designs (Supplementary text 6). Therefore, a hierarchical POM-PTFE covered on a metal substrate is expected to be an ideal dual-selective thermal emitter with a high solar reflectance.

Based on the above molecular and microstructural designs, a hierarchical POM-PTFE bead-like fibrous film coated on an Al foil substrate was synthesized as an ideal dual-selective thermal emitter (denoted as dual-selective POM-PTFE-Al) via a scalable roll-to-roll electrospinning process (Fig. 2c–e, Supplementary Figs. 9, 10 and Supplementary Movie 1). The Al foil substrate (~15 µm-thick, Supplementary Fig. 11) was used primarily as a heat reflector to eliminate the ambient heat transmitted through the polymer film. The POM-PTFE film was carefully tailored (~150 µm-thick, $m_{(POM)}/m_{(PTFE)}$ 7/3) and a disordered stack of bead-like fibers consisting of nano-sized POM fibers (0.3–1.0 µm, center at ~500 nm) and micro-sized PTFE particles (1.0–3.0 µm, center at ~1.8 µm) was formed (Fig. 2e, inset of Fig. 2f, Supplementary texts 7, 8, and Supplementary Figs. 12–18). The appropriate size distribution close to the wavelength range of the solar spectrum, the disordered arrangement of the fibers, and the highly reflective Al foil render the POM-PTFE-Al thermal emitter with a high solar reflectance of up to 95.4% in the 0.3–2.5 µm waveband (Fig. 2f and Supplementary Fig. 19). Furthermore, in the 4–25 µm MIR region, the POM-PTFE-Al thermal emitter exhibits a highly selective emittance in both atmospheric windows (83.2% in 8–13 µm and 67.5% in 16–25 µm, respectively, Fig. 2f) and a high reflectance in the remaining non-window wavebands (Supplementary Fig. 20), showing a desired dual-selective characteristic. Moreover, the dual-selective emitter also exhibits a high tensile strength of 8.2 MPa (Supplementary Fig. 21 and Supplementary Table 3). Besides, its spectral responses show little or almost no changes during and after the stability tests, including 300 h of intense UV irradiation (125 W m$^{-2}$, equivalent to ~1253 days of continuous outdoor exposure in Beijing, Fig. 2g, Supplementary text 9, and Supplementary Fig. 22), 5 months of outdoor exposure in Beijing (Fig. 2g and Supplementary Fig. 23), and 12 h of continuous water treatment (Supplementary Figs. 24 and 25 and Supplementary Movie. 2), clearly indicating its high mechanical strength, high UV resistance, high outdoor stability, and high waterproofness.

These results demonstrate that the synthesized POM-PTFE-Al exhibits ideal characteristics of the dual-selective RC model and high

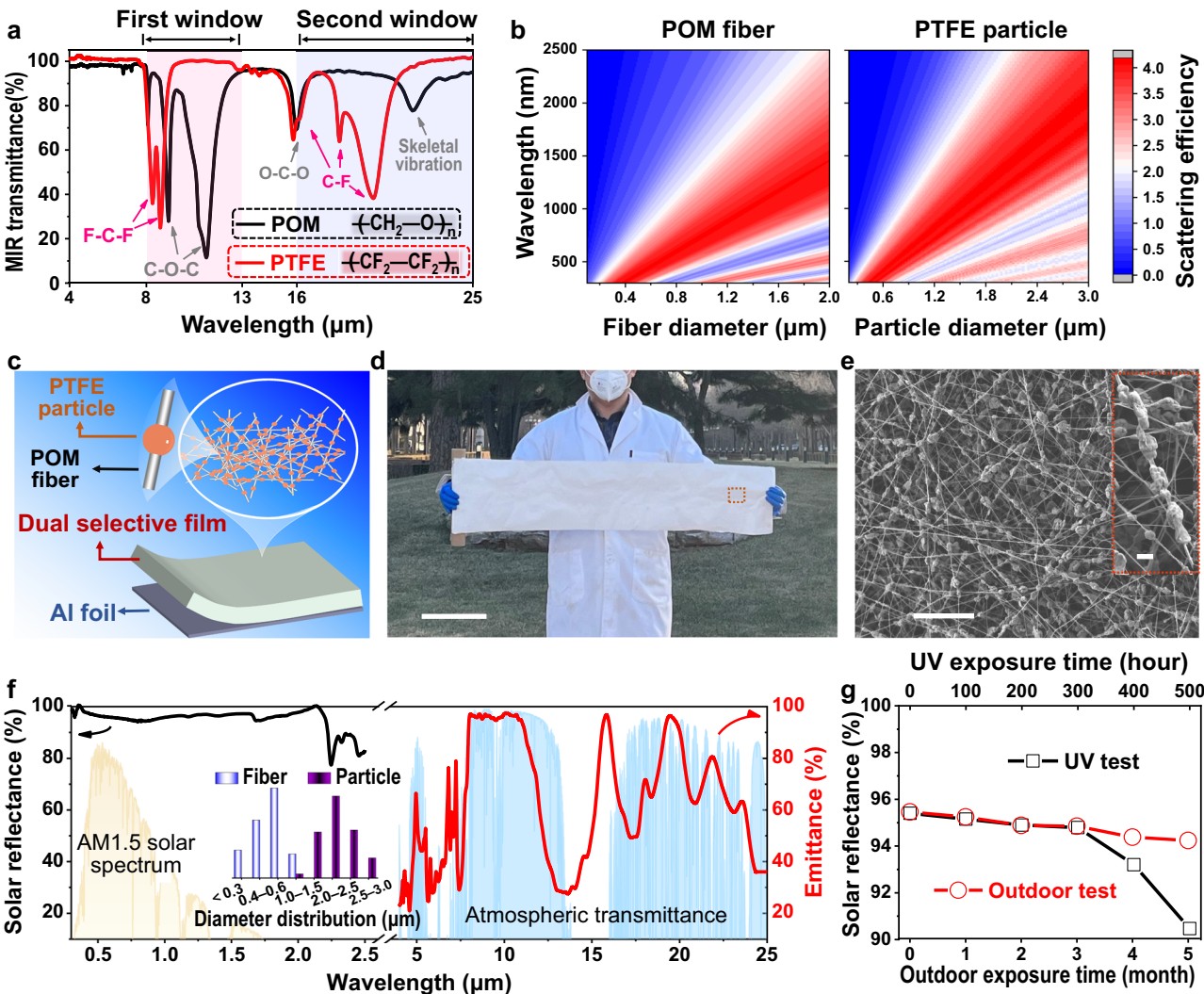

**Fig. 2 | Design, preparation, and spectral analysis of the dual-selective POM-PTFE-Al emitter. a** Molecular design. FTIR-ATR spectra of POM and PTFE where the main characteristic peaks of C-O-C vibrational absorption/emission of POM are located in the region of the first atmospheric window (8–13 μm), and the C-F vibrational absorption/emission of PTFE mainly in the second atmospheric window (16–25 μm). The red and blue boxes indicate the first and second atmospheric window, respectively. **b** Hierarchical nano-micro structure design. Simulation of the scattering efficiency of POM fibers (left) and PTFE particles (right) over the wavelength range of 0.3–2.5 μm with the fiber diameter varied from 0.2 to 2 μm and the particle diameter varied from 0.2 to 3 μm. **c** Schematic of the dual-selective POM-

PTFE emitter. **d**, **e** Photograph (**d** scale bar = 20 cm) and SEM images **e** of the synthesized POM-PTFE film, which has a bead-like fiber structure consisting of POM nanofibers and PTFE micro-particles. Scale bars in **e** = 20 μm and 2 μm (inset). **f** Spectral response of a POM-PTFE film covered Al foil (POM-PTFE-Al) in the 0.3–25 μm waveband, including the solar reflectance (black line) and MIR emittance (red line). The inset shows the statistical distribution of the diameters of the POM nanofibers and PTFE particles inside the POM-PTFE film. **g** Spectral response of POM-PTFE-Al film before and after intense UV and outdoor exposure tests. Source data are provided as a Source Data file.

stability, suggesting the potential to provide a better cooling performance than the existing non-selective and mono-selective thermal emitters, especially in arid climates.

## Thermal measurements

The subambient cooling performance and the net cooling power of the dual-selective POM-PTFE-Al were measured in the hot and perennially dry Ulan Buh desert (in Inner Mongolia, Northwest China, 106°48′93″ E, 40°43′69″ N), where such an arid environment was chosen to maximize the cooling potential of the second atmospheric window (16–25 μm). The specially designed measuring devices were used for the subambient cooling performance test (Fig. 3a, b) and each device was mainly consisted of a polymer film, a wind shield (PE film), surrounding insulating foam covered with Al foil (to shield the surrounding heat), and a carefully calibrated K-type thermocouple (to monitor the sample temperature in real time). Field tests showed that on a hot and dry

sunny day (RH ~ 10%, $T_{ambient} > 30$ °C, and solar > 600 W m$^{-2}$ and peak at ~860 W m$^{-2}$), the dual-selective POM-PTFE-Al achieved a subambient temperature reduction ($T_{ambient} - T_{sample}$) of up to ~9 °C, and ~12 °C subambient cooling on a clear night with a higher humidity (RH = ~45%) (Fig. 3c, d and Supplementary Figs. 26, 27), demonstrating its high subambient cooling performance in real scenarios.

Besides, a POM film covered on Al foil with mono-selective RC properties (donated as mono-selective POM-Al) and a polyvinylidene fluoride (PVDF) film with non-selective RC properties (donated as non-selective PVDF) were also employed for a comparison with the dual-selective sample (Fig. 3e, Supplementary text. 10 and Supplementary Figs. 8a and 28–30). As mentioned above, the strong molecular bonding vibrations of POM are only concentrated in the first atmospheric window of 8–13 μm waveband (Fig. 2a), whereas the molecular bonding vibrations of PVDF are distributed throughout the MIR waveband (including the non-window MIR region, Supplementary

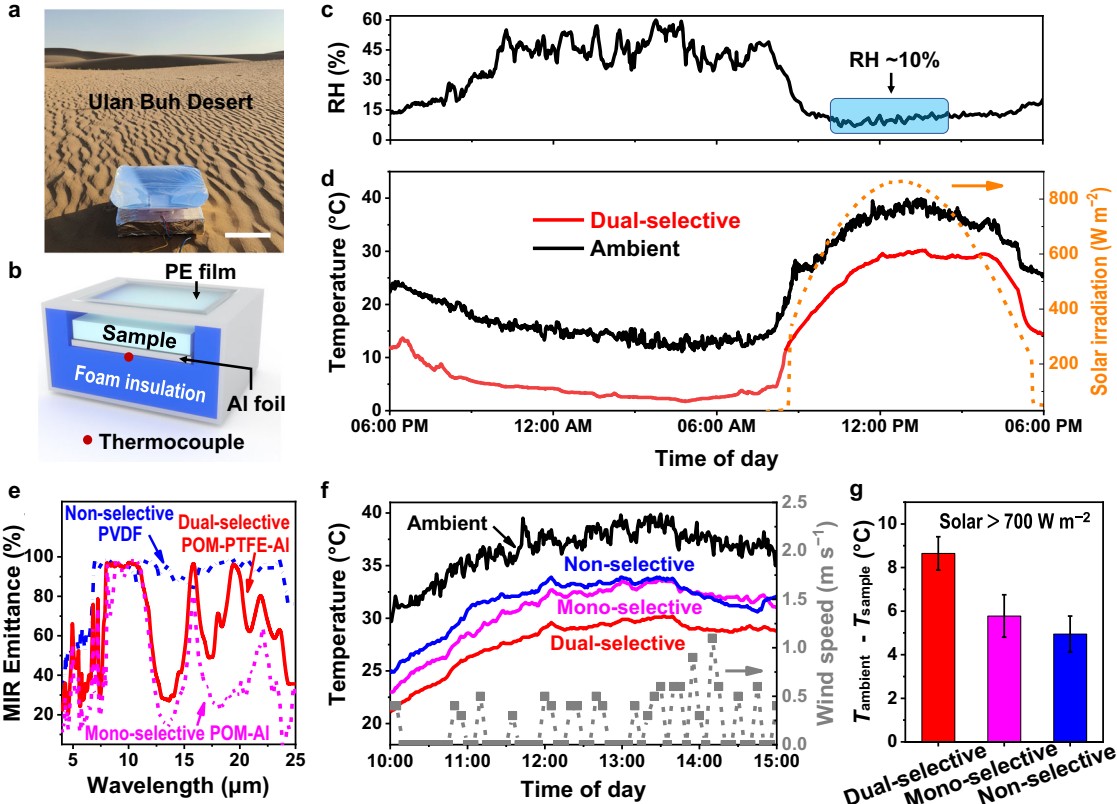

**Fig. 3 | Subambient cooling measurement of the dual-selective POM-PTFE-Al emitter in the Ulan Buh Desert (in Inner Mongolia, China, 6 and 7 September 2022). a, b** Photograph **a** and schematic **b** of the thermal measurement device. Scale bar = 20 cm. **c, d** Continuous measurement of RH **c** and temperature **d** over 24 h, along with the solar irradiance to characterize the subambient cooling performance of the dual-selective POM-PTFE emitter. The blue box in **c** indicates an ultralow RH environment. **e** Comparison of the MIR emittance of the dual-selective POM-PTFE-Al, mono-selective POM-Al, and non-selective PVDF. **f, g** Real-time temperature measurements **f** and the corresponding average subambient temperature reduction **g** of the three different thermal emitters from 10:00 a.m. to 3:00 p.m. (7 September 2022). Error bars in **g** indicate measurement variations of the samples at different times (measured at one-minute intervals over 5 h) and show the mean ± sd (*n* = 301). Source data are provided as a Source Data file.

Fig. 29), which are prerequisites for the mono- and non-selective RC materials, respectively. After a similar electrospinning process and further careful engineering, the as-synthesized POM nanofiber film (~120 μm, Supplementary Figs. 8a and 28) covered on Al foil exhibited a high selective emissivity in the first atmospheric window (Fig. 3e, 77.3% in the 8–13 μm waveband), while the PVDF film (~300 μm, Supplementary Fig. 30) showed a high emittance throughout the MIR waveband (Fig. 3e, ~90% in the 4–25 μm waveband). Besides, both of them had a high solar reflectance of nearly 95% in the 0.3–2.5 μm waveband (Supplementary Fig. 31), which is close to that of the dual-selective POM-PTFE-Al, conforming to the mono-selective and non-selective RC models, respectively.

As shown in Fig. 3f, g, in the hottest and driest time interval of a day (RH = ~10%, $T_{ambient}$ = ~37 °C) under strong sunlight (700–900 W m$^{-2}$) from 10:00 a.m. to 3:00 p.m., the dual-selective POM-PTFE-Al exhibits the highest subambient temperature reduction (8.7 ± 0.8 °C) among the three samples, which is notably higher than that of the non-selective PVDF (5.0 ± 0.8 °C) and the mono-selective POM (5.8 ± 1.0 °C). The higher subambient daytime cooling capacity of the dual-selective sample is mainly due to the selective MIR spectrum compared with the non-selective PVDF and the higher selective emittance in the second atmospheric window compared with the mono-selective POM. It is noteworthy that the cooling performance of the mono-selective POM is also clearly superior to that of the non-selective PVDF at a relatively low ambient wind speed from 10:00 a.m. to 12:00 a.m. (Fig. 3f and Supplementary Fig. 32),

which is also because of its selective MIR spectral property that allows a better cooling performance than the non-selective sample in a low non-radiative heat (near ideal) environment. However, the superiorities disappear when the ambient wind speed increases (1:00 p.m. to 3:00 p.m.) because of the increase in the non-radiative heat around the sample caused by the increased wind speed. The increased non-radiative heat will notably reduce the cooling performance of the mono-selective RC materials, as their net cooling power is much lower than that of non-selective materials.

The net cooling power of dual-selective emitters was also measured in a near-constant ambient environment (at midday from 11:30 a.m. to 1:30 p.m., $T_{ambient}$ = 37.8 °C, RH = ~10%, and solar ~840 W m$^{-2}$, Supplementary Fig. 33a). The as-employed measurement device is similar to the subambient cooling test device described above but with the addition of a smart heater to maintain the same temperature between the sample and ambient air in real time during the measurement (Fig. 4a). It was found that even under strong solar irradiation, the dual-selective POM-PTFE-Al still exhibited an high net cooling power (151.8 ± 13.1 W m$^{-2}$), which was well above that of the mono-selective POM (87.9 ± 9.4 W m$^{-2}$) (Fig. 4b). Taking the cooling power offset by solar heating (~40 W m$^{-2}$) into consideration, the dual-selective sample in such a dry environment had a cooling power close to the theoretical limit of the dual-selective model (~208 W m$^{-2}$, Supplementary Fig. 33b). In comparison, the dual-selective sample was shown to have a cooling power close to the non-selective one (Supplementary text 11 and Supplementary Figs. 34 and 35). These results

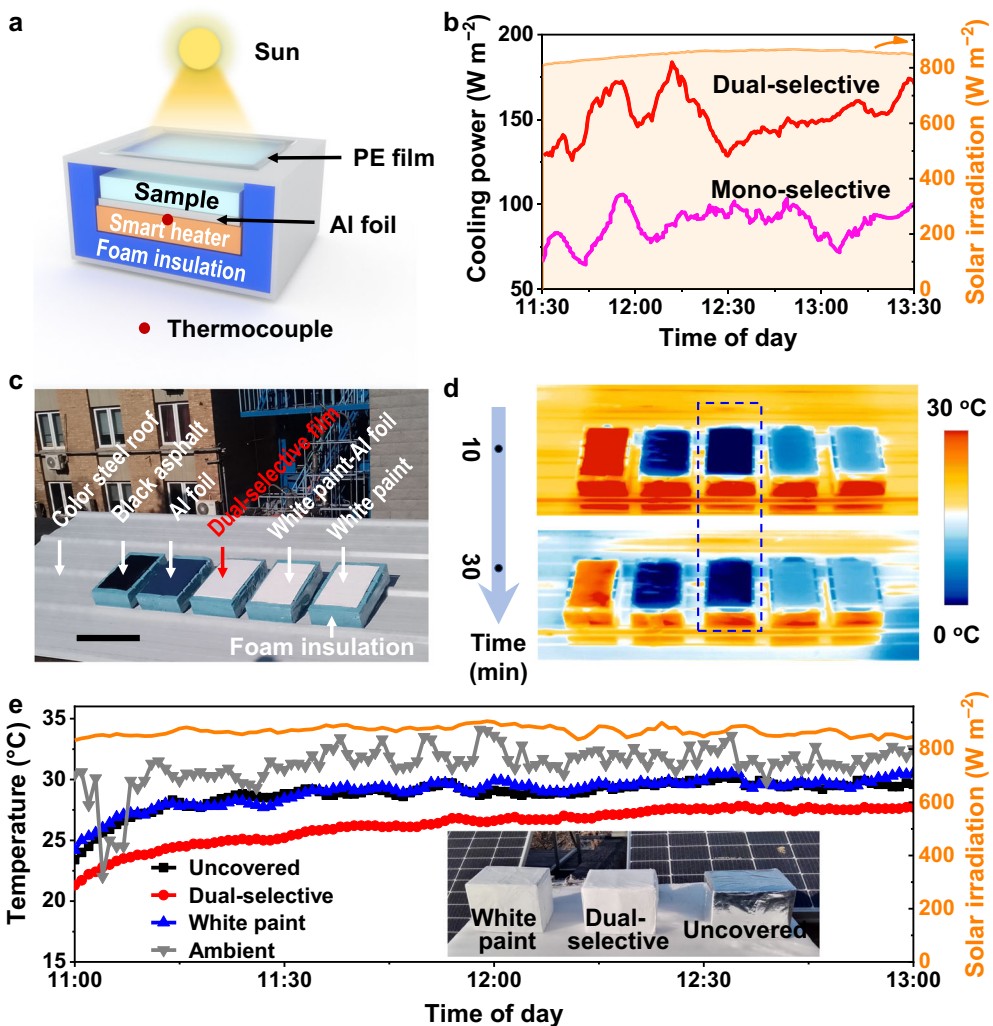

**Fig. 4 | Cooling power and outdoor cooling performance measurements of the dual-selective emitter. a** Schematic for the thermal measurement device used to measure the cooling power of samples. **b** Real-time cooling power measurements of the dual-selective POM-PTFE-Al and mono-selective POM-Al from 11:30 a.m. to 01:30 p.m. (in the Ulan Buh Desert, RH = ~10%, 7 September 2022). The orange region in **b** indicates the solar irradiation. **c**, **d** Infrared test of the dual-selective thermal emitter in an arid outdoor environment (RH = ~10%, 31 January 2023) of Beijing, China (40°0'33" N, 116°20'0.6" E). The photograph **c** and infrared images (**d** 30 min) of the dual-selective sample and several typical commercial roofing materials, including color steel roof (white, background), uncovered black asphalt (10 × 20 cm²), and covered black asphalt (Al foil-covered, dual-selective film covered, white paint-Al foil covered, and white paint-coated). Scale bar in **c** = 20 cm. It should be noted that the infrared tests are qualitative, as the colors of an infrared image depend not only on the surface temperature but also on the MIR emittance. The emittance of each sample in the infrared images was not calibrated individually, which is obviously inappropriate for the ultralow emissivity Al foil sample. **e** Real-time internal temperature measurements (RH = ~10%, Beijing, China, 29 March 2023) of the dual-selective film-covered Al foil box (20 × 15 × 15 cm³) as a proof-of-concept, compared to the uncovered and white paint-coated Al foil boxes (Inset is the photograph of the Al foil boxes for temperature measurement). Source data are provided as a Source Data file.

indicate that the as-designed dual-selective thermal emitter has an enhanced daytime radiative cooling performance over the existing mono-selective and non-selective thermal emitters in real scenarios, especially in arid climates, which is in good agreement with the theoretical analysis (Supplementary text 12).

To further evaluate the cooling performance of dual-selective thermal emitters in practical applications (as building materials, for example), several typical commercial roofing materials were employed for a comparison with the dual-selective POM-PTFE-Al on a dry and sunny day (RH = ~10%, Solar = ~900 W m⁻²), including black asphalt, color steel plate (white), white paint, and bare Al foil. As shown in Fig. 4c and Supplementary Figs. 36 and 37, the dual-selective POM-PTFE-Al and white paint were covered/coated on the surface of a commercial black asphalt (10 × 20 cm²) and placed together with an uncovered black asphalt, an Al foil, and a corrugated color steel plate (0.8 × 2.4 m², as background) on an open roof. Their surface

temperature was visually presented and recorded using an infrared camera (Testo). The results showed that the dual-selective sample had the lowest surface temperature among these samples, and was approximately 33°, 19°, 10°, and 9 °C lower than that of the uncovered black asphalt, color steel roof (white), white paint, and white paint coated Al foil, respectively (Fig. 4d, Supplementary Figs. 38 and 39 and Supplementary Movie 3), qualitatively indicating a higher cooling capacity than these commercial roofing materials.

It is worth noting that the cooling performance between the Al foil and the dual-selective samples cannot be judged from the infrared images alone. This is because the Al foil sample, a good thermal reflector, has a much lower thermal emittance than the other roof samples (the colors of an infrared image depend on both the surface temperature and the thermal emittance). With this in mind, a building-like model experiment was also carried out using an Al-foil box (Al foil covered cardboard box, 0.6 cm-thick) (Fig. 4e). Specifically, three Al

foil boxes ($20 \times 15 \times 15\ cm^3$, uncovered, covered with dual-selective sample and coated with white paint, respectively) were placed in a dry and sunny outdoor scenario and the temperatures inside the boxes were measured in real time using thermocouples (RH = ~13%, Solar ~900 W m$^{-2}$, Supplementary Figs. 40 and 41). As a result, compared with the bare Al foil box and the white painted box, the temperature inside the box covered with the dual-selective sample decreases by 2–5 °C (Fig. 4e and Supplementary Fig. 42), which implies an expected saving of up to ~20–50% of the total building energy consumption[56–58], indicating its superiority as an energy-efficient building material. These results demonstrate the high practicality of dual-selective thermal emitter in real environments.

### Color compatibility

In addition to the need for cooling, RC materials are often required to be multi-functional in practical applications, e.g., they are expected to be multi-colored for esthetic demands. The color of most previously reported daytime RC materials is usually monotonous white or silver. The addition of conventional colored dyes usually leads to a notable weakening or even loss of their cooling effect due to the absorption in the visible waveband, which is a key factor limiting their applications[26,27]. Dual-selective emitters offer improved cooling performance over existing mono-selective and non-selective emitters (especially in arid climates) and are supposed to have better multi-function compatibility, including color compatibility.

As a proof-of-concept, a facile tri-layer design was used to colorize the dual-selective POM-PTFE-Al (marked as colored dual-selective emitter), which was prepared by directly covering the colored polyethylene (PE) film on the surface of the POM-PTFE-Al (Fig. 5a). The translucent colored PE films (~15 μm, Fig. 5b and Supplementary Figs. 43 and 44) are commercially available, inexpensive, harmless, transparent to heat, and can improve the tensile strength of colored thermal emitters (Supplementary Fig. 21 and Supplementary Table 3). PE films can be easily attached to the dual-selective films by hot pressing or spot welding[10,56,59]. The cooling mechanism is that the colored PE film (top) provides the desired color (reflects visible light in a specific waveband), while the dual-selective film (bottom) provides a high reflectance in the near-infrared (NIR) waveband and the dual-selective characteristics in the MIR waveband (Fig. 5a). Clearly, the desired color and size of the colored dual-selective emitter can be easily achieved with this strategy. Figure 5c shows three large sizes of colored dual-selective emitters ($0.2 \times 1\ m^2$) in red, yellow, and blue, respectively. Although the color compatibility inevitably leads to some loss of cooling performance due to the reduction in solar reflectance ($R_{solar}$), theoretical calculations show that subambient cooling can still be achieved even with a low $R_{solar}$ of 80% for a dual-selective thermal emitter under strong sunlight (800 W m$^{-2}$) (Fig. 5d and Supplementary Fig. 45), indicating a high color compatibility.

Taking the colored dual-selective emitters based on three primary colors (red, yellow and blue, respectively, Supplementary Fig. 46) as examples, all three colored emitters have high NIR reflectance (> 94%, Fig. 5e, f) and still retain their dual-selective properties in the MIR waveband (Supplementary Fig. 47), indicating that the three-layer design is feasible to achieve colorful radiative cooling. In addition, the solar reflectance of the red, yellow and blue samples was 85.9%, 84.4%, and 64.7% respectively (Fig. 5f). Among these samples, the solar reflectance of the red and yellow ones was more than 80%, which was expected to achieve a subambient cooling and was demonstrated by a thermal measurement (Fig. 5g). Specifically, on a dry and sunny day (RH = ~15%, Solar = ~800 W m$^{-2}$, and $T_{ambient}$ = ~22 °C) in Beijing, the red and yellow samples showed a subambient temperature reduction of 5.0° and 3.3 °C, respectively (Fig. 5h, i and Supplementary Figs. 48 and 49).

Furthermore, the IR camera tests in a similar environment (RH ~10%, Solar ~700 W m$^{-2}$, and $T_{ambient}$ ~16 °C, Fig. 5j (left) and

Supplementary Figs. 50 and 51) showed that the yellow dual-selective emitter covered on black asphalt had a surface temperature of ~7 °C lower than the yellow PE covered white painted asphalt (right of Fig. 5j and Supplementary Movie 4), indicating the high cooling performance of colored dual-selective emitters. The same conclusion can be drawn from the IR measurement of a blue dual-selective sample (Supplementary text 13, Supplementary Figs. 52 and 53, and Supplementary Movie 5). These results demonstrate the high color compatibility of dual-selective POM-PTFE-Al, and therefore a high practicality in real scenarios.

## Discussion

We demonstrated a dual-selective RC model to exploit the cooling potential of RC materials as it has notably better cooling performance than the existing mono-selective and non-selective models in real conditions, especially in arid climates. As a proof-of-concept, a scalable dual-selective POM-PTFE-Al thermal emitter was fabricated based on the above model. It exhibits a highly selective emittance in both the first (8–13 μm, 83.2%) and second (16–25 μm, 67.5%) atmospheric windows, a high reflectance in the remaining non-window MIR wavebands, a high solar reflectance (0.3–2.5 μm, 95.4%), and a high durability. Due to the desired dual-selective properties, the dual-selective POM-PTFE-Al thermal emitter exhibits a superior daytime subambient cooling capacity over that of existing typical thermal emitters (≥ 3 °C cooler) under strong sunlight (700–900 W m$^{-2}$). The dual-selective sample-based roofs showed superior cooling effect over that of the common commercial roofs. Furthermore, when compatible with multi-color, the subambient cooling can still be achieved by the colored samples. This work provides a scalable design for the reliable passive cooling in real conditions, paving the way for the next-generation energy-saving thermal management.

## Methods

### Fabrication of POM-PTFE, POM, and PVDF electrospun films

For POM-PTFE electrospun film, 5 wt% POM solution of 1,1,1,3,3,3-hexafluoro-2-propanol (HFIP, 99%, Aladdin) was first prepared by adding POM powder (commercial grade, Aladdin) to HFIP solvent and then stirring continuously for 1 h at 60 °C. PTFE micron-particles (1–3 μm, DUPONT) were then added to the POM solution and a homogeneous suspension was formed by sonication (1 h) and continuous stirring (overnight). Subsequently, the mixture solution was electrospun using a 20–gauge needle tip at a voltage of 15 kV, a feeding rate of 4 ml h$^{-1}$, and a spinning distance of 18 cm. The relative humidity and temperature during spinning were maintained below 55% and 25 °C, respectively. The resulting electrospun POM-PTFE film (~150 μm, a POM to PTFE mass ratio of 7:3) was covered on an Al foil and used as a dual-selective thermal emitter. The POM films were obtained by the same method without the addition of PTFE micron particles. The PVDF films were prepared using the previously reported electrospinning method. Specifically, PVDF powder (Mv ~370000, Kynar) was added to a mixed solvent (4:1, v/v) of dimethylformamide (DMF, 99%, Aladdin) and acetone (AR, Tongguang) to produce a 13 wt% PVDF solution after continuous stirring at 50 °C for 5 h. The resulting mixture was electrospun using a 19-gauge needle tip at a voltage of 12 kV, a feeding rate of 0.5 ml h$^{-1}$, a spinning distance of 20 cm, a relative humidity of <40%, and a spinning temperature of 32 °C.

### Morphology characterization

Optical images of the samples were taken using an Honor phone (Play5T). The microstructure of the thermal emitters was characterized by scanning electron microscopy (JSM7401F, JEOL Ltd., Japan).

### Spectral characterization

The spectral responses of the thermal emitters in the solar spectrum (0.3–2.5 μm) and MIR (2.5–25 μm) wavebands were characterized

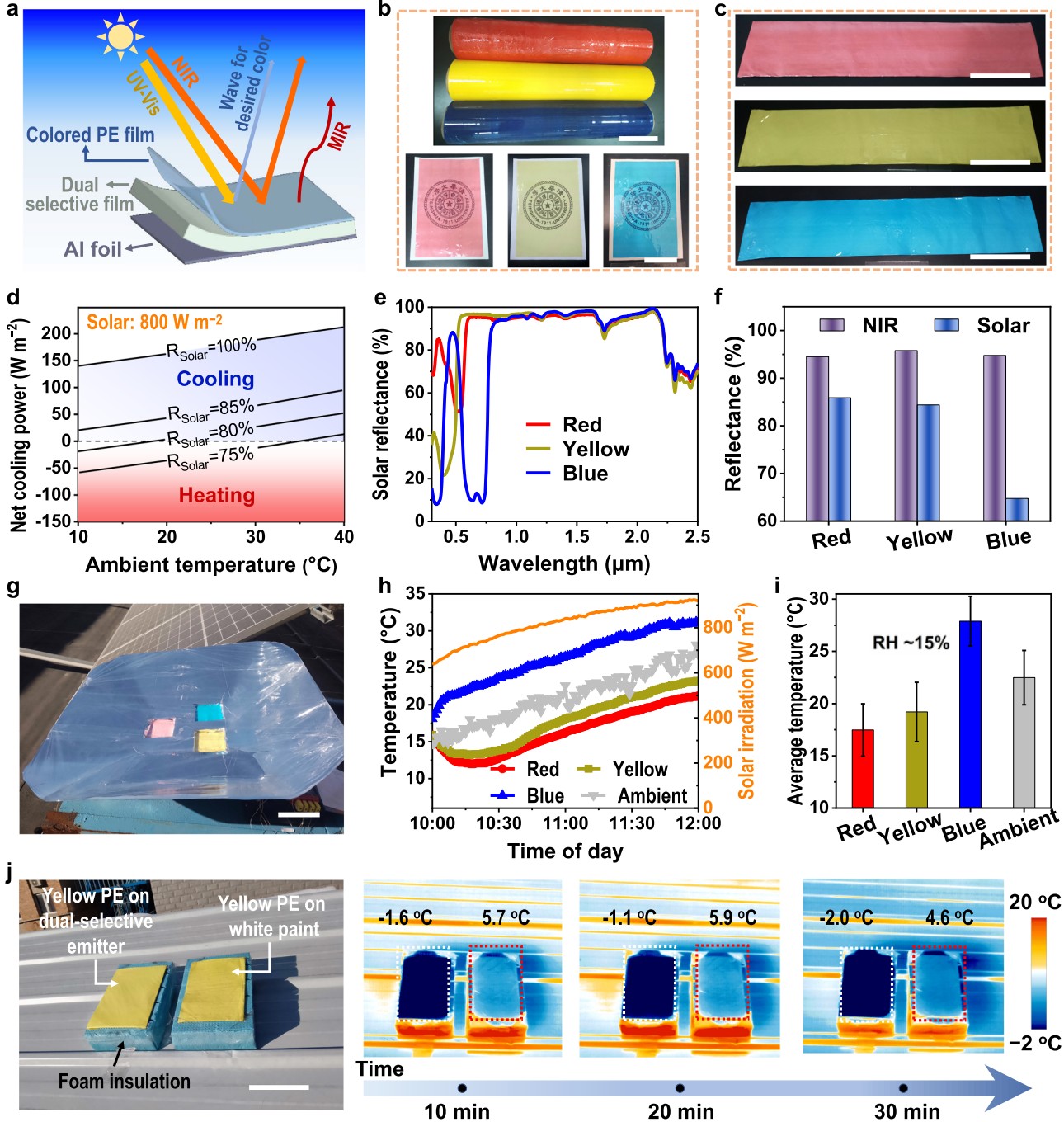

**Fig. 5 | Color compatibility of the dual-selective POM-PTFE-Al. a** Cooling mechanism of the colored dual-selective thermal emitter, consisting of the commercial-colored PE (top) and the dual-selective film (bottom). The colored PE selectively reflects visible light to obtain desired color. The dual-selective film covered on Al foil reflects any sunlight transmitted by the top layer and radiates heat out. **b, c** photograph of commercial-colored PE (**b** scale bars = 10 cm (upper) and 5 cm (lower)) and bilayer designed colored thermal emitters (**c** red (top), yellow (middle), and blue (bottom), scale bar = 20 cm). **d** Theoretical cooling power of a dual-selective thermal emitter as a function of ambient temperature for different $R_{solar}$. Subambient cooling can be achieved even with a low $R_{solar}$ of 80% for a dual-selective cooler, indicating a high color compatibility. **e, f** The solar reflectance spectra **e** and the corresponding average reflectance **f** of red, yellow, and blue

cooler, respectively. **g–i** Thermal measurement of the colored POM-PTFE-Al emitter in Beijing, China (RH = -15%, 31 January 2023). Specifically, photograph of the thermal measurement device (**g**, scale bar = 10 cm); Real-time temperature measurements **h** and the corresponding average temperature **i** of the three different colored thermal emitters from 10:00 a.m. to 12:00 a.m. **j** Actual outdoor cooling performance test for the yellow dual-selective thermal emitter covered asphalt in Beijing, China (RH = -10%, 31 January 2023) compared to the yellow PE film covered white painted asphalt, including their photograph (left) and IR images (right). Scale bar = 10 cm. Error bars in **i** indicate measurement variations of the samples at different times (measured at one-minute intervals from 10:30 a.m. to 12:00 a.m.) and show the mean ± sd (*n* = 91). Source data are provided as a Source Data file.

separately. The former was recorded using an ultraviolet-visible-near-infrared (UV-vis-NIR) spectrophotometer (Cary 7000, Agilent) equipped with an integrating sphere model (Internal DRA-2500, Agilent, barium sulphate as baseline material). The latter was recorded using a

Fourier transform infrared spectrometer (FTIR, INVENIO, Bruker) equipped with a gold integrating sphere (A562, Bruker, gold as baseline material) and an attenuated total reflection module (ATR, diamond as baseline material).

## Solar reflectance and MIR emittance calculation

The equation for calculating the average solar reflectance ($R_{solar}$, 0.3–2.5 μm) of the sample is

$$R_{solar} = \frac{\int_{0.3}^{2.5} I_{solar}(\lambda) R(\lambda) d\lambda}{\int_{0.3}^{2.5} I_{solar}(\lambda) d\lambda} \quad (1)$$

where $I_{solar}(\lambda)$ is the ASTMG173−03 AM1.5 Global Tilt spectrum. $R(\lambda)$ is the measured spectral reflectance of the sample. Similarly, the equation for calculating the average emittance in first atmospheric window ($\varepsilon_1$, 8–13 μm) of the sample is

$$\varepsilon_1 = \frac{\int_{8}^{13} I_{bb}(T,\lambda) \varepsilon(\lambda) d\lambda}{\int_{8}^{13} I_{bb}(T,\lambda) d\lambda} \quad (2)$$

the average emittance in the second atmospheric window ($\varepsilon_2$, 16–25 μm) is

$$\varepsilon_2 = \frac{\int_{16}^{25} I_{bb}(T,\lambda) \varepsilon(\lambda) d\lambda}{\int_{16}^{25} I_{bb}(T,\lambda) d\lambda} \quad (3)$$

where $I_{bb}(T,\lambda)$ and $\varepsilon(\lambda)$ are the blackbody radiation intensity and the measured spectral emittance of the sample, respectively ($\lambda$ is wavelength). The $I_{bb}(T,\lambda)$ can be canculated from

$$I_{bb}(T,\lambda) = \frac{4\pi c^2 \hbar}{\lambda^5} \frac{1}{e^{\frac{2\pi\hbar c}{\lambda k_B T}} - 1} \quad (4)$$

where $c$, $\hbar$, and $k_B$ are the speed of light ($3\times10^8 \, m\,s^{-1}$), the reduced Planck constant ($1.055\times10^{-34} \, J\,s$), and Boltzmann constant ($1.381\times10^{-23} \, J\,K^{-1}$), respectively.

## UV exposure test

The intense UV exposure performance was evaluated by exposing the dual-selective POM-PTFE film in a UV light source (Philips, 125 W m⁻²). The spectral response was tested after every 100 hours of continuous intense UV exposure.

## Mechanical test

The tensile strength of the samples (2 cm wide, 10 cm long) was tested by a Servo tensile testing machine (HZ-1004A) using a gauge with a distance of 6 cm and a displacement rate of 10 mm min⁻¹.

## Water resistance test

The dual-selective POM-PTFE film ($4 \times 4$ cm²) was first floated on non-flowing tap water (in a circular tank, 10 cm-diameter) for 6 h, then the water was made to flow by magnetic stirring for another 6 h. The waterproofness of the sample was assessed by the visual and spectral measurements before and after the water treatment. The water contact angle of the sample was also measured using a contact angle analyzer (XG-CAMA1) in a constant environment (temperature 25 °C, humidity 45%).

## Outdoor exposure test

The outdoor long-term durability test at real-world condition was conducted by continuously exposing the dual-selective POM-PTFE film in an outdoor open roof of Tsinghua university (Beijing, China) with direct sky exposure for 5 months (September, 2022 to February, 2023). The spectral response was tested after every one month of continuous outdoor exposure.

## Thermal measurements

**Subambient cooling performance test**. The subambient cooling effects of the different samples were evaluated using the devices consisting of a sample, aluminum foil, surrounding insulating foam, and a K-type thermocouple (as shown in Fig. 3a, b). The aluminum foil and foam in the test devices were used to minimize the surrounding thermal effect, which is a common method used in previous work. The K-type thermocouples were utilized to monitor the real-time temperature of the sample, which were carefully calibrated to ensure that the difference between measured temperatures was mainly caused by the difference between the samples. The real-time environmental conditions (including relative humidity, wind speed, and ambient temperature) during the thermal measurements were measured and recorded by a weather station (NK5500, Kestrel) located adjacent to the test devices. The input solar power was measured and recorded by a datalogging solar power meter (TES-1333R) located adjacent to the test devices.

**Net cooling power test**. The net cooling power of the samples was measured using a device similar to that used in the thermal test above, with the addition of a smart heater system (as shown in Fig. 4a). The system consists mainly of a DC power supply (ARRAY, 3645 A), a Kapton heater, a power meter (Chroma, 66205), two temperature controllers (Omron), and K-type thermocouples. The system was able to regulate the heating power in real time (heated by a DC source powered Kapton heater) to ensure the same temperature between the sample and the ambient air, and the corresponding heating power was recorded by a digital power meter. This heating power is equal to the cooling power of the sample.

## Real outdoor cooling performance measurements

**For the dual-selective thermal emitter**. The actual outdoor cooling performance of the dual-selective thermal emitter was tested in dry outdoor environment (RH = -10%) of Beijing, China (an open roof of Tsinghua University, 40°0′33″ N, 116°20′0.6″ E). Several typical commercial roofing materials were used for comparison with the dual-selective sample, including color steel roof (white, as background), uncovered black asphalt ($10 \times 20$ cm²), and covered black asphalt (Al foil covered, dual-selective film covered, white paint-Al covered, and white paint-coated, respectively). The surface temperatures of these samples were monitored using an infrared camera (Testo 890, 31 January 2023). The internal temperatures of the uncovered, dual-selective film-covered, and white paint-coated Al foil boxes ($20 \times 15 \times 15$ cm³) were recorded using K-type thermocouples (29 March 2023).

**For the colored dual-selective sample**. A yellow PE film was covered on a white paint coated commercial black asphalt ($10 \times 20$ cm²) and it was employed for comparison with a yellow dual-selective film covered on a black asphalt of the same size. Their surface temperatures were monitored using the same infrared camera as the above test on an open roof of Tsinghua University (31 January 2023). In addition, a similar infrared measurement was performed for a blue dual-selective film in a similar environmental condition for proof-of-concept (8 October 2023).

## Reporting summary

Further information on research design is available in the Nature Portfolio Reporting Summary linked to this article.

## Data availability

Source data are provided in this paper. The source Data file has been deposited in Figshare under the accession link (https://doi.org/10.6084/m9.figshare.23686743). Source data are provided in this paper.

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

## Acknowledgements

R.Z. acknowledges the support from the National Key Research and Development Program of China (2020YFA0210702, 2020YFC2201103), the Tongcheng R&D Foundation, the Tsinghua-Toyota Joint Research Fund, and the National Natural Science Foundation of China (22075163, 51872156, 62134009, 62121005). W.L. acknowledges the support from the National Natural Science Foundation of China (62134009 and 62121005).

## Author contributions

R.Z., Xue.W. conceived the idea. Xue.W. designed the models and experiments. Xue.W., F.X. and W.L. performed the modeling work. Xue.W. performed the material preparation and characterization with the help of J.Z., J.L., Y.Zhang, Xun.W., S.Zhao, Q.J., S.Zhang, B.W., Y.L., D.G., R.L., F.W., Y.H. and Y.Zhao. Xue.W. wrote the manuscript. R.Z. supervised the project. All the authors provided discussion and comments.

## Competing interests

The authors declare no competing interests.
