## [Peer Review File · Nature Communications]

REVIEWER COMMENTS

Reviewer #1 (Remarks to the Author):

This manuscript focuses on the development of radiative cooling materials, showcasing results that include the longer wavelength region of the atmospheric window, which is generally not considered. Using a mass-production-capable method based on electrospinning and a mixed solution with particles, large-area radiative cooling materials were produced. Experimental validation demonstrated high emissivity in the dual-band region. Temperature measurements in practical setting confirmed that the dual-band design offers superior cooling performance. Additionally, the combination with colored transparent films demonstrates its potential as a colored radiative cooler. While there is sufficient data to support the concept, it is essential to faithfully address the following comments provided by the reviewers, given the high reputation of Nature Communications.

- The dual-selective concept is not being introduced for the first time. Therefore, there is a need to enhance the introduction by providing more comprehensive references and background information.
- The authors mentioned that humidity has a significant impact on the second atmospheric window band. While the results under conditions with RH below 10% are well-presented in the experiments, it's worth considering that changes in the atmospheric window due to humidity can significantly affect the cooling flux of the radiative cooler. Adding comments on the cooling performance based on changes in the atmospheric window due to humidity and utilizing humidity-dependent atmospheric window data could provide valuable insights.
- When the particle diameter changes, it would be interesting to know how the experimental conditions of electrospinning vary. While adjusting the thickness of POM fiber can be relatively easy through factors like solution concentration, distance, voltage, etc., providing process information related to changes in particle size would be valuable.
- Could the authors provide more detailed information about the experimental conditions related to the fabrication? For example, the amount of HFIP used when preparing a 5 wt% POM solution, the quantity of POM powder used in grams, or the total amount of POM-PTFE solution prepared for the fabrication of samples with a large area.
- Include the equation for the calculation of net cooling power also.
- (Optional) In studies involving nanostructures that exhibit colors, researchers typically provide color data through CIE 1931. It might be beneficial to calculate and provide results based on the measured spectrum of the radiative cooling material, demonstrating a good match with experimental data. This can be relatively straightforward to do and could be worth considering.

Reviewer #2 (Remarks to the Author):

In this study, the authors presented an impressive approach to subambient radiative cooling devices. They devised a dual-selective radiative cooling (RC) system, which targeted not only the primary atmospheric transparency window but also an additional window spanning from 16 to 25 micrometers. Their findings revealed the superior cooling performance of this dual-selective design when compared to non-selective and mono-selective counterparts. Furthermore, the authors demonstrated the compatibility of this innovative strategy with colored materials like colored polyethylene. If the issues listed below can be adequately revised, it could be suitable for publication in Nature Communications.

1. The manuscript needs better organization to enhance reader understanding and readability. The author should carefully review and correct minor editing errors throughout the manuscript. For instance, in line 103, (d) should be corrected to (b), and in lines 347 and 475, "Figure" should consistently be noted as "Fig" for uniformity. Moreover, the supplementary information contains a significant amount of redundant data. For instance, Figure S3 resembles Figure 1e, differing only by two data points. It is advisable to replace Figure 2a with Figure S14. Additionally, Figure S33 (a, b) is already included in Figure 4, while Figure S35 replicates the content of Figure 4e inset. Both Figure S38 and S42 are duplicates of information already present in Figure 5. Currently, the excessive redundancy and lack of organization notably diminish the readability of this article.

2. In Figure 1, the authors argue that the dual-selective RC theoretically outperforms other counterparts, relying on their "theoretical limit of sub-ambient temperature reduction" as a basis for this argument. However, this argument needs a more comprehensive and clearly articulated explanation. For instance, it would be beneficial for the author to elucidate the specific points they intend to convey with Figure 1c. Currently, there is a lack of detailed explanations in this regard. Also, the net cooling power of non-selective RC should be also present to Figure 1d to avoid any bias. Furthermore, it is essential to address why, in the calculations for net cooling power, the non-selective RC system appears to exhibit superior cooling power compared to the others. This discrepancy between net cooling power and theoretical limit needs clarification. A more in-depth exploration of these aspects will enhance the clarity and comprehensibility of the paper's findings.

3. In Figure 1e, the author presents the simulated "theoretical limit of subambient temperature reduction," which serves as the primary supporting data for their central argument. To effectively convince the readers, it is crucial that this data is thoroughly explained. The authors have offered an explanation for why, in ideal scenarios, mono-selective RC outperforms non-selective RC, but they have yet to provide a similar explanation for dual-selective RC. It is imperative that they elucidate the reasons why, in real-world applications, the dual-selective RC system surpasses both the non-selective and mono-selective counterparts. Additionally, an explanation should be provided for the scenario in which mono-selective RC outperforms dual-selective RC in an ideal context.

4. In Figure 2e, the author notes that they have determined fiber and particle sizes based on scattering efficiency simulations. To bolster this claim, the author should consider including a comparison with the reflection data of a neat Al foil in Figure 2f.

5. The author should detail the quantitative method employed for characterizing fiber and particle sizes. This should encompass information regarding the number of samples characterized and averaged (not relative), as well as the specific software or methodologies employed in the analysis.

6. In Figure 4b, the author presents the cooling power of both dual-selective and mono-selective RC systems based on real experiments. To ensure a fair comparison and maintain consistency throughout the manuscript, it would be advisable to include the cooling power data for the non-selective RC system as well. Additionally, it would be valuable if the author could explore potential correlations between the simulated cooling powers and the measured cooling powers, providing further insights into the accuracy and reliability of the simulations in predicting real-world performance.

7. The author highlights that the dual-selective radiative cooling (RC) system exhibits better color compatibility. However, it's worth noting that for the demonstrated colored RC, the cooling performance does not appear to be superior. In particular, the blue-colored RC seems to exhibit heating rather than cooling, likely due to its high absorbance in the 0.6-0.8 μm range. It might not be entirely appropriate to claim better color compatibility if the demonstrated RC systems are not achieving actual cooling, especially considering the existence of research papers showcasing effective cooling performance with colored materials. To strengthen their argument, the author should provide additional data or experimental results that demonstrate the actual cooling behavior of the blue-colored RC.

Response Letter to Reviewers

Response to Reviewer #1

This manuscript focuses on the development of radiative cooling materials, showcasing results that include the longer wavelength region of the atmospheric window, which is generally not considered. Using a mass-production-capable method based on electrospinning and a mixed solution with particles, large-area radiative cooling materials were produced. Experimental validation demonstrated high emissivity in the dual-band region. Temperature measurements in practical setting confirmed that the dual-band design offers superior cooling performance. Additionally, the combination with colored transparent films demonstrates its potential as a colored radiative cooler. While there is sufficient data to support the concept, it is essential to faithfully address the following comments provided by the reviewers, given the high reputation of Nature Communications.

Reply: Thank you very much for a careful evaluation of our manuscript. We really appreciate your positive comments and constructive suggestions, which are very helpful for us to improve the quality of this manuscript.

Comment 1. The dual-selective concept is not being introduced for the first time. Therefore, there is a need to enhance the introduction by providing more comprehensive references and background information.

Reply: Thank you for your valuable suggestion. We agree with you that dual-selective concept is not being introduced for the first time, and it is also known as dual emission or dual band (*Angew. Chem. Int. Ed.* 60, 22624–22638, (2021); *Nat. Photonics* 13, 277–283, (2019)). However, the existing dual-selective concept is quite different from our dual-selective design in this work. Based on your suggestion, we have added more comprehensive background information and necessary references in the revised manuscript to distinguish our dual-selective design from the previous one. The details are as follows.

Existing dual-selective (or dual-emission) materials are mainly used in bioimaging (*Nat. Methods* 20, 1563–1572, (2023); *Nat. Commun.* 11, 4, (2020); *Nat. Photonics* 13, 277–283, (2019)), sensing (*Chem. Soc. Rev.* 49, 5110–5139, (2020); *Sensors* 16, 820, (2016)), thermophotovoltaic cell (*Phys. Rev. Lett.* 107, 045901, (2011); *Opt. Express* 19, 15221–15228, (2011)), and information security/protection (*Adv. Mater.* 35, 2300177, (2023)), which usually show selective emission in two

different wavelengths or narrow wavebands and low emission in the remaining detected wavebands.

Unlike these existing dual-emission materials, our dual-selective design refers to the high selective emission in the two atmospheric window wavebands (8–13 μm and 16–25 μm , respectively) to use the two atmospheric windows to achieve efficient heat dissipation from the Earth to the cold outer space, which has not been studied before. Moreover, in addition to the above-mentioned difference in the dual-selective emission properties, our dual-selective design also needs to achieve the high reflectance in both the non-windowed MIR and solar wavebands, which is an essential difference from the existing dual-band materials. Therefore, our dual-selective design is different from the existing concept. There is no such dual-selective design in the field of daytime radiative cooling yet.

The added references (refers 1–5) are shown as follows.

- [1] Jiang, L. et al. Large Stokes shift fluorescent RNAs for dual-emission fluorescence and bioluminescence imaging in live cells. *Nat. Methods* 20, 1563–1572, (2023).
- [2] Tang, X., Ackerman, M. M., Chen, M. L. & Guyot-Sionnest, P. Dual-band infrared imaging using stacked colloidal quantum dot photodiodes. *Nat. Photonics* 13, 277–283, (2019).
- [3] Behera, S. K., Park, S. Y. & Gierschner, J. Dual Emission: Classes, Mechanisms, and Conditions. *Angew. Chem. Int. Ed.* 60, 22624–22638, (2021).
- [4] Liu, X. et al. Taming the Blackbody with Infrared Metamaterials as Selective Thermal Emitters. *Phys. Rev. Lett.* 107, 045901, (2011).
- [5] Zheng, H. Q. et al. Photo-Stimuli-Responsive Dual-Emitting Luminescence of a Spiropyran-Encapsulated Metal-Organic Framework for Dynamic Information Encryption. *Adv. Mater.* 35, 2300177, (2023).

Based on your suggestions, we have added the relevant revisions into the main text (Page 6) and references (Page 30) sections of the revised manuscript as follows (shown in *red texts*).

“However, despite the existence of various dual-selective designs³⁷⁻⁴¹, no such desired dual-selective materials have yet been experimentally developed before. It remains a challenge to design a dual-selective thermal emitter through a simple method that has precise spectral control of the two atmospheric windows, non-window MIR, and solar wavebands.” (Page 6 of the revised manuscript)

References

37 Behera, S. K., Park, S. Y. & Gierschner, J. Dual Emission: Classes, Mechanisms, and Conditions. *Angew. Chem. Int. Ed.* 60, 22624–22638, (2021).

38 Jiang, L. et al. Large Stokes shift fluorescent RNAs for dual-emission fluorescence and bioluminescence imaging in live cells. *Nat. Methods* 20, 1563–1572, (2023).

39 Liu, X. et al. Taming the Blackbody with Infrared Metamaterials as Selective Thermal Emitters. *Phys. Rev. Lett.* 107, 045901, (2011).

40 Tang, X., Ackerman, M. M., Chen, M. L. & Guyot-Sionnest, P. Dual-band infrared imaging using stacked colloidal quantum dot photodiodes. *Nat. Photonics* 13, 277–283, (2019).

41 Zheng, H. Q. et al. Photo-Stimuli-Responsive Dual-Emitting Luminescence of a Spiropyran-Encapsulated Metal-Organic Framework for Dynamic Information Encryption. *Adv. Mater.* 35, 2300177, (2023)." (Page 31 of the revised manuscript)

Comment 2. The authors mentioned that humidity has a significant impact on the second atmospheric window band. While the results under conditions with RH below 10% are well-presented in the experiments, it's worth considering that changes in the atmospheric window due to humidity can significantly affect the cooling flux of the radiative cooler. Adding comments on the cooling performance based on changes in the atmospheric window due to humidity and utilizing humidity-dependent atmospheric window data could provide valuable insights.

Reply: Thank you for your suggestion. According to your suggestion, we have added the additional simulations and corresponding analyzes of the humidity-induced cooling performance of three different types of thermal emitters (dual-, mono-, and non-selective radiative cooling (RC) models). The different atmospheric window data based on humidity were used for the simulations. The details are as follows.

To evaluate the effect of humidity on the radiative cooling performance of dual-, mono-, and non-selective thermal emitters, we also simulated their cooling performance in a higher humidity environment with different non-radiative heat effects ($h = 0, 1, 2, 4, 6 \text{ W m}^{-2} \text{ K}^{-1}$), in addition to the arid (low humidity) environment. The atmospheric transparency data in a high humidity environment for the simulation were the same with the "Sub-humid" data in Fig. 1b in the revised manuscript. As

can be seen in **Fig. R1-1**, the atmospheric transparency in the second atmospheric window waveband (16–25 μm) is much reduced compared with the low humidity data due to the strong heat absorption by water in this waveband.

The theoretical calculations show that in a high humidity environment, the mono-selective RC model shows optimal subambient cooling performance in both ideal ($h = 0 \text{ W m}^{-2} \text{ K}^{-1}$) and real ($h \geq 1 \text{ W m}^{-2} \text{ K}^{-1}$) scenarios (**Fig. R1-2**), which is very different from the cooling performance in low humidity conditions (**Fig. R1-3**). More importantly, **in real scenarios, the dual-selective RC model shows a subambient cooling performance close to the mono-selective model (especially for $h = 2\text{--}6 \text{ W m}^{-2} \text{ K}^{-1}$) and better than the non-selective model (1.3–7.8 $^{\circ}\text{C}$ cooler) (Fig. R1-2)**, which results from its higher cooling power (71.8–136.5 W m^{-2} , $T_{\text{ambient}} = 0\text{--}40 \text{ }^{\circ}\text{C}$) than the mono-selective RC model (67.8–130.6 W m^{-2}) (**Fig. R1-4**) and less atmospheric parasitic heat than the non-selective RC model.

Fig. R1-1. Atmospheric transparency data of different humidities used for the simulations. (Supplementary Fig. 7a in the revised Supplementary Materials)

Fig. R1-2. Theoretical subambient cooling performance of the three different radiative coolers in a high humidity environment. (Supplementary Figs. 6 and 7b in the revised Supplementary Materials)

Furthermore, compared with the low humidity environment, the net cooling power and the subambient temperature reduction in the high humidity environment are notably reduced (Fig. R1-5), as expected, due to the fact that the atmospheric transparency of the atmosphere is notably reduced as the ambient humidity increases, resulting in a low heat dissipation efficiency. It is worth noting that the cooling performance decrease of the mono-selective RC model with increasing humidity is significantly smaller than that of the other two RC models. Specifically, the cooling power reduction of the mono-selective model is only 12.9–25.1 $W m^{-2}$ ($T_{ambient}$, 0–40 $^{\circ}C$), whereas the reductions for the dual-selective and non-selective RC models reach 49.1–77.7 $W m^{-2}$ and 59.6–102.9 $W m^{-2}$, respectively (Fig. R1-5a). Moreover, in real scenarios ($h = 1–6 W m^{-2} K^{-1}$), the decrease of the subambient cooling temperature of the mono-selective model is only 2–8 $^{\circ}C$, whereas the cooling temperature reduction of the other two RC models reaches 7–20 $^{\circ}C$ (Fig. R1-5b-5d). This is due to the fact that with the humidity increase, the reduction in the atmospheric transparency in the main atmospheric window (8–13 μm) is much less than the reduction in the second window (16–25 μm) (Fig. R1-1). Therefore, the degradation in cooling performance of the mono-selective model is much less than that of the dual- and non-selective models.

These results show that with the ambient humidity increase, the cooling performance of all

three thermal emitters decreases accordingly, but the dual-selective emitter still maintains a subambient cooling performance close to the mono-selective thermal emitter and notably better than the non-selective thermal emitter. Importantly, with the ambient humidity decrease, the cooling performance increase of the dual-selective thermal emitter is significantly larger than that of the mono-selective thermal emitter, indicating that the former has a better subambient cooling performance than the latter in real scenarios.

Fig. R1-3. Theoretical subambient cooling performance of the three different radiative coolers in a low humidity environment. (Fig. 1e in the revised manuscript)

Fig. R1-4. Comparison of the theoretical cooling power of the three different radiative coolers at different T_{ambient} (-20° to 40°C). (Supplementary Fig. 6f in the revised Supplementary Materials)

Fig. R1-5. a–d, Comparison of cooling power (a) and subambient temperature reduction of the three different radiative coolers (b–d) between low humidity and higher humidity. (Supplementary Fig. 7f in the revised Supplementary Materials)

The corresponding revisions have been added to the revised Manuscript (Page 6) and revised Supplementary Materials (Supplementary text 4, Supplementary Figs. 6,7) as follow.

“The simulations in a higher humidity environment show that even in a high-humidity environment the dual-selective model still maintains a subambient cooling performance close to the mono-selective model and notably better than the non-selective model (Supplementary text 4 and Supplementary Figs. 5–7).” (Page 6 of the revised manuscript)

“Supplementary text 4. Modelling of the effect of humidity on the radiative cooling performance of non-selective, mono-selective, and dual-selective RC models

To evaluate the effect of humidity on the radiative cooling performance of dual-, mono-, and non-selective RC models, in addition to the arid (low humidity) environment, we also simulated their cooling performance in a higher humidity environment (with different non-radiative heat effects, $h = 0, 1, 2, 4, 6 W m^{-2} K^{-1}$, Supplementary Figs. 6,7).

The atmospheric transparency data in a high humidity environment for the simulation were the same with the "Sub-humid" data in Fig. 1b, as shown in Supplementary Fig. 7a. As can be seen, the atmospheric transparency in the second atmospheric window waveband (16–25 μm) is much reduced compared with the low humidity data due to the strong heat absorption by water in this waveband.

The theoretical calculations show that in a high humidity environment, the mono-selective RC model shows optimal subambient cooling performance in both ideal ($h = 0 \text{ W m}^{-2} \text{ K}^{-1}$) and real ($h \geq 1 \text{ W m}^{-2} \text{ K}^{-1}$) scenarios (Supplementary Figs. 6a–e,7b), which is very different from the cooling performance in low humidity conditions (Fig. 1e). More importantly, in real scenarios, the dual-selective model shows a subambient cooling performance close to the mono-selective model (especially for $h = 2\text{--}6 \text{ W m}^{-2} \text{ K}^{-1}$) and better than the non-selective model (1.3–7.8 $^{\circ}\text{C}$ cooler) (Supplementary Fig. 7b), which results from its higher cooling power (71.8–136.5 W m^{-2} , $T_{\text{ambient}} = 0\text{--}40 \text{ }^{\circ}\text{C}$) than the mono-selective model (67.8–130.6 W m^{-2}) (Supplementary Fig. 6f) and less atmospheric parasitic heat than the non-selective model.

Furthermore, compared with the low humidity environment, the net cooling power and subambient temperature reduction are notably reduced in the high humidity environment (Supplementary Figs. 7c–f) due to the fact that the atmospheric transparency of the atmosphere is notably reduced as the ambient humidity increases, resulting in a low heat dissipation efficiency. It is worth noting that the cooling performance decrease of the mono-selective RC model with increasing humidity is significantly smaller than that of the other two RC models. Specifically, the cooling power reduction of the mono-selective model is only 12.9–25.1 W m^{-2} ($T_{\text{ambient}}, 0\text{--}40 \text{ }^{\circ}\text{C}$), whereas the reductions for the dual-selective and non-selective models reach 49.1–77.7 W m^{-2} and 59.6–102.9 W m^{-2} , respectively (Supplementary Fig. 7c). Moreover, in real scenarios ($h = 1\text{--}6 \text{ W m}^{-2} \text{ K}^{-1}$), the decrease of the subambient cooling temperature of the mono-selective model is only 2–8 $^{\circ}\text{C}$, whereas the cooling temperature reduction of the other two RC models reaches 7–20 $^{\circ}\text{C}$ (Supplementary Figs. 7d–f). This is due to the fact that with the humidity increase, the reduction in atmospheric transparency in the main atmospheric window (8–13 μm) is much less than the reduction in the second window (16–25 μm) (Supplementary

Fig. 7a). Therefore, the degradation in cooling performance of the mono-selective model is much less than that of the dual- and non-selective models.

These results show that with the ambient humidity increase, the cooling performance of all three RC models decreases accordingly, but the dual-selective model still maintains a subambient cooling performance close to the mono-selective RC model and notably better than the non-selective RC model. Importantly, with the ambient humidity decrease, the cooling performance increase of the dual-selective RC model is significantly larger than that of the mono-selective RC model, indicating that the former has a better subambient cooling performance than the latter in real scenarios.” (Pages 10,11 in the revised Supplementary Materials)

Supplementary Fig. 6. Theoretical cooling performance of the three different radiative coolers in a high humidity environment. a-e, Cooling power of the three different radiative coolers as a function of the emitter temperature in an arid environment ($T_{\text{ambient}} = 20 \text{ }^\circ\text{C}$). f, Comparison of the theoretical cooling power of the three different radiative coolers at different T_{ambient} (-20° to $40 \text{ }^\circ\text{C}$). The calculation is based on the same atmospheric transmittance data with the "sub-humid" in Supplementary Fig. 5.” (Page 28 in the revised Supplementary Materials)

Supplementary Fig. 7. Supplementary Fig. 7. Comparison of cooling performance of the three different radiative coolers between low humidity and higher humidity. a, Atmospheric transmittance data of low and high humidity environments. b, Theoretical limit of subambient cooling of the three types of radiative coolers at different h (0–6 $W m^{-2} K^{-1}$). The calculation is based on the atmospheric transmittance data in high humidity, a net cooling power P_{net} of 0 $W m^{-2}$ and $T_{ambient} = 20$ °C. c–f, Comparison of cooling power (c) and subambient temperature reduction of the three different radiative coolers (d–f) between low humidity and higher humidity.” (Page 29 in the revised Supplementary Materials)

Comment 3. When the particle diameter changes, it would be interesting to know how the experimental conditions of electrospinning vary. While adjusting the thickness of POM fiber can be relatively easy through factors like solution concentration, distance, voltage, etc., providing process information related to changes in particle size would be valuable.

Reply: Thank you for your valuable suggestion. To answer your questions, we have added the detailed design principle for PTFE particle size and also clarified the effect of changes in PTFE particle size on the electrospinning process following your suggestion, as followings.

1. Design principles of the PTFE particle size

According to the Mie scattering theory, particle size is indeed the key to achieving high solar reflectance for a particle-based daytime radiative cooling material. We have chosen the appropriate particle size distribution of PTFE particles according to the following principles.

1) First, according to our theoretical calculations for PTFE particles based on Mie theory (**Fig. 2b** in the revised manuscript), for a given thickness of a radiative cooler which is fixed, the highest solar reflectance can be achieved in an air medium when the size distribution of PTFE particles is around 0.2–3.0 μm (close to the solar waveband). **It should be noted that the result was obtained for PTFE particles in an air medium where there is a large difference in refractive indices between PTFE and air, resulting in strong scattering at the polymer/air interface.** However, **for the POM-PTFE complexes-based samples, considering the matching of POM fibers and PTFE particles in the samples, we prefer to select PTFE particles with a size larger than that of POM fibers.** The reason is that as-used polymers, *i.e.*, PTFE and POM, both have similar refractive indices (~ 1.5 , *Astrophys Space Sci* 39, L13-L18, (1976); *Mon. Not. R. Astron. Soc.* 175, 197-207, (1976); *Optics InfoBase Conference Papers*, (2007)). **If the size of the PTFE particle is similar or smaller than that of the POM fiber (e.g., using nano-sized PTFE particles to prepare the POM-PTFE electrospun film), the Mie scattering effect of the particles will be masked by the POM fibers because the PTFE nanoparticles are embedded in the POM fibers (i.e., the PTFE particles are not in the air medium).** Therefore, the PTFE particles with a larger size distribution than that of the POM fibers were used to prepare the dual-selective samples.

2) In addition, our previous work has shown that nano-sized POM fibers with diameter distribution close to the main waveband of sunlight (0.2–1.0 μm) can be easily produced by electrospinning (*Nat. Sustain.*, 2023, <https://doi.org/10.1038/s41893-023-01200-x>).

3) Finally, nanoparticle-based products are susceptible to health risks due to the inevitable inhalation of the human body during production and use. Besides, although both nano- and micron-sized PTFE particles are commercially available, nano-sized PTFE particles are much more expensive than micron-sized PTFE particles.

Therefore, taking the Mie scattering effect, compatibility with POM fibers, safety, and cost of PTFE particles into account, the POM fibers with distribution of 0.2–1.0 μm and PTFE particles with distribution of 1.0–3.0 μm were used to prepare the dual-selective samples.

2. Effect of changes in PTFE particle size on the electrospinning process

As analyzed above, the PTFE particles with smaller sizes are not ideal for this work. Nevertheless, it is meaningful to qualitatively discuss the effect of PTFE particle size on the electrospinning process. According to the Mie scattering theory, if the size of the PTFE particles is changed (in this work it refers to reducing the size of the PTFE particles from the micro size distribution to the nano size distribution), it is necessary to prepare POM fibers with larger diameters (centered around 600 nm, as shown in **Fig. 2b** of the revised manuscript), so that their diameter distribution is closer to the main wavebands of sunlight. The larger diameter of electrospun fibers can be achieved in one or more of the following ways: increasing the concentration of the electrospinning solution, decreasing the spinning voltage, and decreasing the distance between the positive and negative electrodes (*Adv. Mater.* 13, 70–72, (2001); *Macromolecules* 41, 4746–4752, (2008); *Polymers* 15, 65, (2023); *Chem. Rev.* 119, 8, 5298–5415, (2019); *J. Electrostat.* 35, 151–160, (1995)).

Relevant revisions have been added to the revised manuscript (Page 10) and revised Supplementary Materials (Supplementary text 6, Pages 13,14) as follows.

*“As shown in **Fig. 2b**, a strong scattering (high reflectance) in the solar waveband can be achieved when the fiber or particle diameter distribution is close to the solar waveband (200–3000 nm), providing a clear guide for hierarchically structural designs (Supplementary text 6). Therefore, a hierarchical POM-PTFE covered on a metal substrate is expected to be an ideal dual-selective thermal emitter with a high solar reflectance.”* (Page 10 of the revised manuscript)

*“**Supplementary text 6. Design principles for the diameter distribution of PTFE particles and POM fibers***

According to the Mie scattering theory, the diameter distribution of the particles or fibers is the key to achieving a high solar reflectance of a particle- or fiber- based thermal emitter (for daytime radiative cooling). We have selected the appropriate size distribution of PTFE particles and POM fibers according to the following principles.

1) First, according to our theoretical calculations for PTFE particles based on the Mie scattering theory (Fig. 2b), for a given thickness of a radiative cooler, the highest solar reflectance can be achieved in an air medium when the size distribution of PTFE particles is around 0.2–3.0 μm (which is close to the solar waveband). It should be noted that the result was obtained for PTFE particles in an air medium where there is a large

difference in refractive indices between PTFE and air, resulting in strong scattering at the polymer/air interface. However, for the POM-PTFE complexes-based samples, considering the matching of POM fibers and PTFE particles in the samples, we prefer to select PTFE particles with a size larger than that of POM fibers. The reason is that the as-used polymers, i.e., PTFE and POM, both have similar refractive indices (~1.5)^{9,11,12}. If the size of the PTFE particle is similar or smaller than that of the POM fiber (e.g., using nano-sized PTFE particles to prepare the POM-PTFE electrospun film), the Mie scattering effect of the particles will be masked by the POM fibers because the PTFE nanoparticles are embedded in the POM fibers (i.e., the PTFE particles are not in the air medium). Therefore, the PTFE particles with a larger size distribution than that of the POM fibers were used to prepare the dual-selective samples.

2) In addition, our previous work has shown that nano-sized POM fibers with diameter distribution close to the main waveband of sunlight (0.2–1.0 μm) can be easily produced by electrospinning¹³.

3) Finally, nanoparticle-based products are susceptible to health risks due to the inevitable inhalation of the human body during production and use. Besides, although both nano- and micron-sized PTFE particles are commercially available, nano-sized PTFE particles are much more expensive than micron-sized PTFE particles.

Therefore, taking the Mie scattering effect, compatibility with POM fibers, safety, and cost of PTFE particles into account, the POM fibers with distribution of 0.2–1.0 μm and PTFE particles with distribution of 1.0–3.0 μm were used to prepare the dual-selective samples.

As an additional note, when changing the size of the PTFE particles (in this work it refers to reducing the size of the PTFE particles from the micro size distribution to the nano size distribution), it is necessary to prepare the POM fibers with larger diameters according to the Mie scattering theory (centered around 600 nm, Fig. 2b), so that their diameter distribution is closer to the main wavebands of sunlight. The larger diameter of electrospun POM fibers can be achieved in one or more of the following ways: increasing the concentration of the electrospinning solution, decreasing the spinning voltage, and decreasing the distance between the positive and negative electrodes¹⁴⁻¹⁸.” (Pages 13,14

of the revised Supplementary Materials)

Comment 4. Could the authors provide more detailed information about the experimental conditions related to the fabrication? For example, the amount of HFIP used when preparing a 5 wt% POM solution, the quantity of POM powder used in grams, or the total amount of POM-PTFE solution prepared for the fabrication of samples with a large area.

Reply: Thank you for your suggestion. We have added more detailed information on the preparation of the POM-PTFE sample based on your suggestion. Specifically, **the resulting POM-PTFE electrospun film has a POM to PTFE mass ratio of 7:3 and a thickness of ~150 μm . As a reference, the total mass of POM-PTFE mixed solution required to prepare 1 m^2 of POM-PTFE electrospun film is approximately 400 g with a temperature of 25 $^{\circ}\text{C}$ and a relative humidity of 45%.**

It is important to note that the total amount of solution in the actual electrospinning process cannot be strictly determined due to the fact that the ambient environments (such as temperature and humidity, which are also key factors for electrospinning) and the instrumentation status of an unspecifically designed electrospinning process are not likely to be constant. In contrast, the mass concentrations of POM or PTFE in the POM-PTFE sample and the film thickness determine the spectral response and other physicochemical properties of a final electrospun film, which are more representative of the preparation characteristics of the electrospinning process.

The corresponding revisions have been added to the Method section of the revised Manuscript (Pages 23) as follow.

“The resulting electrospun POM-PTFE film (~150 μm , a POM to PTFE mass ratio of 7:3) was covered on an Al foil and used as a dual-selective thermal emitter. As a reference, the total mass of POM-PTFE mixed solution required to prepare 1 m^2 of POM-PTFE electrospun film is approximately 400 g (temperature 25 $^{\circ}\text{C}$, relative humidity 45%).”

Comment 5. Include the equation for the calculation of net cooling power also.

Reply: Thank you for your suggestion. We apologize for our carelessness of not clearly describing where these equations are located. Actually, the detailed equations for the calculation of net cooling power are in the **Supplementary text 1** of the Supplementary Materials as follows.

Based on a simple one-dimensional heat transfer model, the energy balance equation for the thermal emitters combined with radiative and non-radiative thermal transfer is

$$P_{net} = P_{rad} - P_{atm} - P_{sun} - P_{non-rad} \quad (1)$$

, where P_{net} is the net cooling power of the thermal emitter; P_{rad} is the emitted thermal radiation from the thermal emitter; P_{atm} is the absorbed thermal radiation from the atmosphere; P_{sun} is the absorbed thermal radiation from sunlight; $P_{non-rad}$ is the non-radiative heat transfer by thermal conduction and thermal convection.

The P_{rad} and P_{atm} are described by Fourier's law, which give the equations

$$P_{rad} = 2\pi \int_0^{\pi/2} \sin\theta \cos\theta d\theta \int_0^{\infty} \varepsilon I_{bb}(T_{sample}) d\lambda \quad (2)$$

$$P_{atm} = 2\pi \int_0^{\pi/2} \sin\theta \cos\theta d\theta \int_0^{\infty} \varepsilon_{atm} I_{bb}(T_{ambient}) d\lambda \quad (3)$$

, where $T_{ambient}$ and T_{sample} are the temperature of the ambient air and the thermal emitter, respectively. ε_{atm} and ε are the emittance of the ambient air and the thermal emitter, respectively. λ is wavelength. And $I_{bb}(T)$ is the blackbody radiation intensity. The $I_{bb}(T)$ can be calculated from

$$I_{bb}(T) = \frac{4\pi c^2 \hbar}{\lambda^5} \frac{1}{e^{\frac{2\pi \hbar c}{\lambda k_B T}} - 1} \quad (4)$$

, where c , \hbar , and k_B are the speed of light ($3 \times 10^8 \text{ m s}^{-1}$), the reduced Planck constant ($1.055 \times 10^{-34} \text{ J s}$), and Boltzmann constant ($1.381 \times 10^{-23} \text{ J K}^{-1}$), respectively.

The non-radiative heat transfer is described by Fourier's law, which gives the equation

$$P_{non-rad} = h(T_{ambient} - T_{sample}) \quad (5)$$

, where h is the non-radiative heat transfer coefficient.

Base on the Eqns. 1–5, P_{net} of the three models ($P_{sun} = 0 \text{ W m}^{-2}$ for theoretical limit) can be calculated by the following equations.

For the non-selective RC model ($\varepsilon = 1$ in the entire MIR waveband of 4–25 μm):

$$P_{net(\text{non-selective})} = 2\pi \int_0^{\pi/2} \sin\theta \cos\theta d\theta \int_4^{25} I_B(T_{sample}) d\lambda - 2\pi \int_0^{\pi/2} \sin\theta \cos\theta d\theta \int_4^{25} \varepsilon_{atm} I_B(T_{ambient}) d\lambda - h(T_{ambient} - T_{sample}) \quad (6)$$

For the mono-selective RC model ($\varepsilon = 1$ in the first atmospheric window of 8–13 μm waveband

and $\varepsilon = 0$ in the remaining MIR waveband):

$$P_{net(\text{mono-selective})} = 2\pi \int_0^{\pi/2} \sin\theta \cos\theta d\theta \int_8^{13} I_B(T_{\text{sample}}) d\lambda - 2\pi \int_0^{\pi/2} \sin\theta \cos\theta d\theta \int_8^{13} \varepsilon_{\text{atm}} I_B(T_{\text{ambient}}) d\lambda - h(T_{\text{ambient}} - T_{\text{sample}}) \quad (7)$$

For the dual-selective RC model ($\varepsilon = 1$ in the two atmospheric windows of 8–13 μm and 16–25 μm wavebands and $\varepsilon = 0$ in the non-window MIR wavebands):

$$P_{net} \quad (\text{dual-selective}) = 2\pi \int_0^{\pi/2} \sin\theta \cos\theta d\theta [\int_8^{13} I_B(T_{\text{sample}}) d\lambda + \int_{16}^{25} I_B(T_{\text{sample}}) d\lambda] - 2\pi \int_0^{\pi/2} \sin\theta \cos\theta d\theta [\int_8^{13} \varepsilon_{\text{atm}} I_B(T_{\text{ambient}}) d\lambda + \int_{16}^{25} \varepsilon_{\text{atm}} I_B(T_{\text{ambient}}) d\lambda] - h(T_{\text{ambient}} - T_{\text{sample}}) \quad (8)$$

The ε_{atm} is described by

$$\varepsilon_{\text{atm}} = 1 - t(\lambda)^{1/\cos\theta} \quad (9)$$

, where $t(\lambda)$ is the atmospheric transparency obtained from the ATRAN modelling software with a water vapor column of 1.0 mm and zero zenith angle.

The theoretical P_{net} of these three different RC models as a function of T_{sample} at a T_{ambient} of 20 °C was obtained when using different h values (0–8 $\text{W m}^{-2} \text{K}^{-1}$) (Fig. 1c and Supplementary Fig. 2), by which we can also obtain the theoretical limits of the subambient temperature reduction of the three models (as the temperature reduction of the thermal radiation reached a steady state when $P_{\text{net}} = 0 \text{ W m}^{-2}$) (Fig. 1e and Supplementary Fig. 3).

Furthermore, the theoretical limits of P_{net} of these three different RC models as a function of T_{ambient} were obtained when $T_{\text{ambient}} = T_{\text{sample}}$ (Fig. 1d and Supplementary Fig. 4), as the non-radiative heat can be excluded ($P_{\text{non-rad}} = h(T_{\text{ambient}} - T_{\text{sample}}) = 0$).

We have added a description of the location of these equations in the revised Manuscript (figure caption of **Fig. 1c**) as follow.

“Fig. 1 | Conceptual explanation and model calculations of dual-selective radiative cooler. c, Theoretical cooling power of these three different radiative coolers as a function of the emitter temperature at a fixed T_{ambient} of 20 °C (for the detailed calculation equations, see Supplementary text 1).....”

Comment 6. (Optional) In studies involving nanostructures that exhibit colors, researchers typically

provide color data through CIE 1931. It might be beneficial to calculate and provide results based on the measured spectrum of the radiative cooling material, demonstrating a good match with experimental data. This can be relatively straightforward to do and could be worth considering.

Reply: Thank you for your suggestion. The CIE chromaticity coordinates of red, yellow, blue dual-selective samples (**Fig. R1-6**) have been added in the revised version as follow.

Fig. R1-6. Commission Internationale de l’Eclairage (CIE) chromaticity coordinates of red/yellow/blue dual-selective films

The corresponding revisions have been added to and the revised manuscript (Page 19) and the revised Supplementary Materials (Supplementary Fig. 46, Page 68) as follows.

“Taking the colored dual-selective emitters based on three primary colors (red, yellow and blue, respectively, Supplementary Fig. 46) as examples, all three colored emitters have high NIR reflectance (>94%, Fig. 5e,f) and still retain their dual-selective properties in the MIR waveband (Supplementary Fig. 47), indicating that the three-layer design is feasible to achieve colorful radiative cooling.” (Page 19 of the revised manuscript)

Supplementary Fig. 46. Commission Internationale de l'Eclairage (CIE) chromaticity coordinates of red/yellow/blue dual-selective films. (Page 68 of the revised Supplementary Materials)

Response to Reviewer #2

In this study, the authors presented an impressive approach to subambient radiative cooling devices. They devised a dual-selective radiative cooling (RC) system, which targeted not only the primary atmospheric transparency window but also an additional window spanning from 16 to 25 micrometers. Their findings revealed the superior cooling performance of this dual-selective design when compared to non-selective and mono-selective counterparts. Furthermore, the authors demonstrated the compatibility of this innovative strategy with colored materials like colored polyethylene. If the issues listed below can be adequately revised, it could be suitable for publication in Nature Communications.

Reply: Thank you very much for your valuable comments and constructive suggestions on our manuscript.

Comment 1. The manuscript needs better organization to enhance reader understanding and readability. The author should carefully review and correct minor editing errors throughout the manuscript. For instance, in line 103, (d) should be corrected to (b), and in lines 347 and 475, "Figure" should consistently be noted as "Fig" for uniformity. Moreover, the supplementary information contains a significant amount of redundant data. For instance, Figure S3 resembles Figure 1e, differing only by two data points. It is advisable to replace Figure 2a with Figure S14. Additionally, Figure S33 (a, b) is already included in Figure 4, while Figure S35 replicates the content of Figure 4e inset. Both Figure S38 and S42 are duplicates of information already present in Figure 5. Currently, the excessive redundancy and lack of organization notably diminish the readability of this article.

Reply: Thank you for your careful checks. Based on your comments, we have carefully reviewed, re-organized, and corrected minor editing errors throughout the manuscript and supplementary materials to improve reader understanding and readability. The details are as follows.

The "**d**" in the line 103 (figure caption of Fig. 1d in the revised manuscript) has been changed to the "**b**". The "Figure" in the line 475 ("Methods-Thermal measurements" section in the revised manuscript, Page 25) has been changed to "Fig". The "Figure" on the line 347 (second paragraph of the "Color compatibility" section of the revised manuscript, Page 19) is the first word of the sentence and is more appropriately used in full, therefore it was retained.

The redundant data in the supplementary information have been removed, including Figs. S3,

S33b, S35, S38, and S42 in the unrevised Supplementary Materials. However, Fig. S33a in the unrevised Supplementary Materials has been retained in the revised version due to the inclusion of the necessary labelling for the samples (from 1# to 6#, Supplementary Fig. 38a, page 60 of the revised Supplementary Materials), which differs from Fig. 4c in the revised manuscript. In addition, both Fig. S14 (Supplementary Fig. 16 in the revised Supplementary Materials) and Fig. 2a have been retained in the revised version. One reason for this is that Fig. 2a was intended to show the molecular design of our dual-selective materials, and did not yet include the preparation of the final sample, so there should be no information about the final POM-PTFE sample, so it has been retained. Another reason is that the data in Fig. S14 were intended to compare the experimentally prepared sample with the previous molecular design, to show that the experimental results and the molecular design were in agreement, which was certainly necessary and not redundant with Fig. 2, and so Fig. S14 has also been retained. To avoid any misunderstanding, we have added a note to the figure caption of Fig. S14 (Supplementary Fig. 16 in revised Supplementary Materials).

Furthermore, we have added the additional data in the revised Supplementary Materials to provide further support for our conclusions (Figs. R2-1 to R2-3), as follows.

Fig. R2-1. Difference in the theoretical temperature reduction compared to the dual-selective radiative cooler at different non-radiative heat transfer coefficients h (1–6 $\text{W m}^{-2} \text{K}^{-1}$) for the mono-selective and non-selective radiative coolers at an ambient temperature of 20 °C. (Supplementary Fig. 3 in the revised Supplementary Materials)

Fig. R2-2. Schematics of the thermal measurements of the building model. (Supplementary Fig. 40 in the revised Supplementary Materials)

Fig. R2-3. Digital image of commercial-colored PE films (red, yellow, and blue, respectively). (Supplementary Fig. 43 in the revised Supplementary Materials)

Relevant revisions have been added to the revised manuscript (figure caption of Fig. 1d) and the revised Supplementary Materials (Supplementary Figs. 3, 16, 38, 40, 43) as follows.

“Fig. 1 | Conceptual explanation and model calculations of dual-selective radiative cooler.d, Comparison of the theoretical cooling power between dual-selective, mono-selective, and non-selective radiative coolers at different $T_{ambient}$ (–20 to 40 °C). The calculation is based on the same atmospheric transmittance with (b) and the same temperature between the thermal emitter T_{sample} and the $T_{ambient}$” (figure caption of Fig. 1d in the revised manuscript)

“The net cooling power of the samples was measured using a device similar to that used in the thermal test above, with the addition of a smart heater system (as shown in Fig. 4a).” (“Methods-Thermal measurements” section in the revised manuscript, Page 25)

Supplementary Fig. 3. Difference in the theoretical temperature reduction compared to the dual-selective radiative cooler at different non-radiative heat transfer coefficients h (1–6 $W m^{-2} K^{-1}$) for the mono-selective and non-selective radiative coolers at an ambient temperature of 20 °C. (Page 25 of the revised Supplementary Materials)

“Supplementary Fig. 16. FTIR-ATR spectra of POM, PTFE and dual-selective POM-PTFE where the main characteristic peaks of C-O-C vibrational absorption/emission of POM are located in the region of the first atmospheric window (8–13 μm), and the C-F vibrational absorption/emission of PTFE mainly in the second atmospheric window (16–25 μm). The figure is intended to compare the experimentally prepared POM-PTFE sample with the previous molecular design, to show that the experimental results and the molecular design are in agreement.” (Page 38 of the revised Supplementary Materials)

Supplementary Fig. 38. Outdoor cooling performance measurements of the dual-selective emitter (Beijing, China, 31 January 2023). a, photograph, and b, the corresponding real-time surface temperatures of the dual-selective emitter and the several typical commercial roofing materials, including color steel roof (white, background, marked as 1#), uncovered black asphalt (10×20 cm², marked as 2#), and covered black asphalt (Al foil-covered 3#, dual-selective film covered 4#, white paint-Al foil covered 5#, and white paint-coated 6#). It should be noted that the infrared tests are qualitative as the colors of an infrared image depend not only on the surface temperature

but also on the MIR emittance of sample materials. In this test, the emissivity of each sample in the infrared images was not calibrated individually and it was assumed to have an emissivity of 0.9, which is obviously inappropriate for the ultra-low emissivity Al foil sample. Thus, the cooling performance between the Al foil and the dual-selective sample cannot be judged from the colors of their infrared images alone." (Supplementary Fig. 38, page 60 of the revised Supplementary Materials)

Supplementary Fig. 40. Schematics of the thermal measurements of the building model." (Page 62 of the revised Supplementary Materials)

Supplementary Fig. 43. Digital image of commercial-colored PE films (red, yellow, and blue, respectively)." (Page 65 of the revised Supplementary Materials)

Comment 2. In Figure 1, the authors argue that the dual-selective RC theoretically outperforms other counterparts, relying on their "theoretical limit of sub-ambient temperature reduction" as a basis for this argument. However, this argue needs a more comprehensive and clearly articulated explanation. For instance, it would be beneficial for the author to elucidate the specific points they intend to convey with Figure 1c. Currently, there is a lack of detailed explanations in this regard. Also, the net cooling power of non-selective RC should be also present to Figure 1d to avoid any bias. Furthermore, it is essential to address why, in the calculations for net cooling power, the non-selective RC system appears to exhibit superior cooling power compared to the others. This discrepancy between net cooling power and theoretical limit needs clarification. A more in-depth exploration of these aspects

will enhance the clarity and comprehensibility of the paper's findings.

Reply: Thank you for your valuable suggestion. Based on your suggestion, we have added the detailed explanations for the Fig. 1c and d, and also added the simulated net cooling power data of non-selective RC to Fig. 1d. The details are as follows.

1. The net cooling power of non-selective RC has been added to the Fig. 1d, as shown in **Fig. R2-4**.

Fig. R2-4. Theoretical cooling power of these three different radiative coolers as a function of the emitter temperature at a fixed T_{ambient} of 20 °C. (Fig. 1d in the revised manuscript)

2. The detailed explanation for the specific points of Fig. 1c and Supplementary Fig. 2.

Specifically, both Fig. 1c and Supplementary Fig. 2 show the curves of cooling power as a function of emitter temperature at a fixed ambient temperature. The theoretical limit of subambient cooling (steady-state temperature) of the sample is reached when the cooling power reaches zero, corresponding to the intersection of the curve with the x -axis in these figures, which is the source of the data in Fig. 1e. The instinctive cooling power of the sample (as shown in Fig. 1d) is reached when there is no temperature difference between the sample and the ambient, corresponding to the intersection of the curve with the y -axis in these figures.

3. Relationship between the net cooling power and the subambient temperature reduction.

The radiative cooling performance of a sample can be expressed in two main ways: the net cooling power and the cooling temperature (temperature reduction or temperature difference), which **are not positively correlated**. The reason for this is that the former represents the rate of cooling (or heat dissipation), which depends on the **"speed" at which heat is transferred from the Earth**

to the cold outer space. The latter is the end result of the heat transfer to the outer space and depends on the total amount of heat transferred to the outer space. Therefore, it is not the case that the higher the cooling power, the greater the cooling temperature reduction. Below is a detailed analysis of the radiative cooling performance (both net cooling power and subambient cooling temperature) for the three typical thermal emitters.

Non-selective thermal emitters can dissipate heat to the outer space across the entire mid-infrared waveband (note that non-window wavebands can also dissipate heat, even if the vast majority of the heat is absorbed/blocked by the atmosphere), whereas mono-selective thermal emitters can only dissipate heat through the narrow atmospheric transparent window of 8–13 μm . Therefore, non-selective thermal emitters have a much faster cooling rate (*i.e.*, higher net cooling power) than mono-selective thermal emitters. As for dual-selective thermal emitters, they can emit heat through both atmospheric transparent windows (8–13 μm and 16–25 μm), much faster than mono-selective thermal emitters and close to non-selective ones. Therefore, dual-selective thermal emitters have a much higher net cooling power than mono-selective and close to non-selective thermal emitters (Fig. 1c).

As analyzed above, while non-selective thermal emitters dissipate heat rapidly (with high cooling power), the trade-off is that most of the heat in the non-window waveband is absorbed/blocked by the atmosphere and returns to re-heat the emitters (*i.e.*, atmospheric parasitic heat). As a result, in ideal scenarios ($h = 0 \text{ W m}^{-2} \text{ K}^{-1}$), non-selective emitters emit the least total amount of heat to the outer space compared to the other two types of thermal emitters, resulting in the smallest temperature reduction ultimately achieved. In contrast, selective thermal emitters can dissipate most of their heat into the outer space, although they cool slowly, but can achieve a much higher temperature reduction once steady state is reached. In addition, as the atmospheric transparency of the second atmospheric window is slightly less than 1 compared with mono-selective emitters, there is also an additional (albeit small) atmospheric heating effect for dual-selective emitters that cannot be ignored under ideal conditions. Therefore, in ideal environments, mono-selective thermal emitters have a slightly higher temperature reduction than dual-selective emitters.

In summary, dual-selective thermal emitters can dissipate heat through both atmospheric windows (*i.e.*, high heat dissipation rate), while avoiding atmospheric parasitic heat (*i.e.*, more heat dissipated into the outer space), thus combining high cooling power with high subambient

temperature reduction. As a result, dual-selective thermal emitters provide enhanced radiative cooling performance than the other two types of emitters in real arid environments.

The corresponding revisions have been added to the revised Manuscript (first paragraph of the “Dual-selective radiative cooling model” section, Fig. 1d, and figure caption of Fig. 1c,d) and revised Supplementary Materials (Supplementary text 2 and figure caption of Supplementary Fig. 2) as follows.

“Fig. 1 | Conceptual explanation and model calculations of dual-selective radiative cooler. c, Theoretical cooling power of these three different radiative coolers as a function of the emitter temperature at a fixed $T_{ambient}$ of 20 °C (for the detailed calculation equations, see Supplementary text 1). The theoretical limit of subambient cooling is reached when the cooling power reaches zero (the intersection of the curve with the x-axis). The calculation is based on the atmospheric transmittance in an arid environment (relative humidity $RH < 2\%$, blue in c) and the non-radiative heat transfer coefficient of $h = 1$ (solid lines) and 4 (dashed lines) $W m^{-2} K^{-1}$. d, Comparison of the theoretical cooling power between dual-selective, mono-selective, and non-selective radiative coolers at different $T_{ambient}$ (-20 to 40 °C). The calculation is based on the same atmospheric transmittance with (b) and the same temperature between the thermal emitter T_{sample} and the $T_{ambient}$” (figure caption of Fig. 1c,d of the revised manuscript)

“In an ideal environment ($h = 0 W m^{-2} K^{-1}$), the dual-selective RC model has a subambient cooling performance close to that of the mono-selective model (~80 °C) and much better than the non-selective model (~50 °C) (Fig. 1c,e, Supplementary text 2 and Supplementary Fig. 2).” (Page 5 of the revised manuscript)

“Supplementary Fig. 2.The theoretical limit of subambient cooling (steady-state temperature) of the sample is reached when the cooling power reaches zero, corresponding to the intersection of the curve with the x-axis in these figures, which is the source of the data in Fig. 1e and Supplementary Fig. 3. The instinctive cooling power of the sample (as shown in Fig. 1d) is reached when there is no temperature difference between the sample and the ambient, corresponding to the intersection of the curve with the y-axis in these figures.” (figure caption of Supplementary Fig. 2, Page 22 of the Supplementary Materials)

“Supplementary text 2. Relationship between the net cooling power and the subambient temperature reduction for dual-selective, mono-selective, and non-selective thermal emitters.

The radiative cooling performance of a sample can be expressed in two main ways: the radiative cooling power and the cooling temperature (temperature reduction or temperature difference), which are not positively correlated. The reason for this is that the former represents the rate of cooling (or heat dissipation), which depends on the "speed" at which heat is transferred from the Earth to the cold outer space. The latter is the end result of the heat transfer to the outer space and depends on the total amount of heat transferred to the outer space. Obviously, high cooling power does not imply large temperature reduction, and our ultimate goal in developing radiative cooling technology is to achieve high temperature reduction in most practical applications. Therefore, we prefer to use the subambient temperature reduction rather than the net cooling power to evaluate the radiative cooling performance of thermal emitters. The following is a detailed analysis of the radiative cooling performance (both net cooling power and subambient cooling temperature) for the three typical thermal emitters.

Non-selective thermal emitters can dissipate heat to the outer space across the entire mid-infrared waveband (note that non-window wavebands can also dissipate heat, even if the vast majority of the heat is absorbed/blocked by the atmosphere), whereas mono-selective thermal emitters can only dissipate heat through the narrow atmospheric transparent window of 8–13 μm . Therefore, non-selective thermal emitters have a much faster cooling rate (i.e., higher net cooling power) than mono-selective thermal emitters.

As for dual-selective thermal emitters, they can emit heat through both atmospheric transparent windows (8–13 μm and 16–25 μm), much faster than mono-selective thermal emitters and close to non-selective ones. Therefore, dual-selective thermal emitters have a much higher net cooling power than mono-selective and close to non-selective thermal emitters (Fig. 1c).

As analyzed above, while non-selective thermal emitters dissipate heat quickly (with high cooling power), the trade-off is that most of the heat in the non-window band is absorbed/blocked by the atmosphere and returns to re-heat the emitters (i.e., atmospheric parasitic heat). As a result, in ideal environments ($h = 0 \text{ W m}^{-2} \text{ K}^{-1}$), non-selective emitters emit the least total amount of heat to the outer space compared to the other two types of thermal emitters, resulting in the smallest temperature reduction ultimately achieved. In contrast, selective thermal emitters can dissipate most of their heat into the outer space, although they cool slowly, but can achieve a much higher temperature reduction once steady state is reached.

In addition, as the atmospheric transparency of the second atmospheric window is slightly less than 1 compared with mono-selective emitters, there is also an additional (albeit small) atmospheric heating effect for dual-selective emitters that cannot be ignored under ideal conditions. Therefore, in ideal environments, mono-selective thermal emitters have a slightly higher temperature reduction than dual-selective emitters.”

(Pages 7,8 of the revised Supplementary Materials)

Comment 3. In Figure 1e, the author presents the simulated "theoretical limit of subambient temperature reduction," which serves as the primary supporting data for their central argument. To effectively convince the readers, it is crucial that this data is thoroughly explained. The authors have offered an explanation for why, in ideal scenarios, mono-selective RC outperforms non-selective RC, but they have yet to provide a similar explanation for dual-selective RC. It is imperative that they elucidate the reasons why, in real-world applications, the dual-selective RC system surpasses both the non-selective and mono-selective counterparts. Additionally, an explanation should be provided for the scenario in which mono-selective RC outperforms dual-selective RC in an ideal context.

Reply: Thank you for your valuable suggestion. We have added detailed explanations for Fig. 1e in the revised version (The effect of non-radiative heat on the subambient cooling performance of the

three types of thermal emitters), as suggested by you. The issues raised in this comment have been addressed point by point. The details are as follows.

As analyzed in the response to your second comment (**Comment 2**), the radiative cooling performance of a sample can be described in two main ways: the radiative cooling power and the cooling temperature (temperature reduction or temperature difference). Since our simulations show that **high cooling power does not imply high temperature reduction**, and our ultimate goal in developing radiative cooling technology is to achieve high temperature reduction in most practical applications. Therefore, **we prefer to use the subambient temperature reduction rather than the net cooling power to evaluate the radiative cooling performance of these thermal emitters.** The following are our responses to the issues raised in the comment relating to subambient temperature reduction.

1. The detailed explanations of the subambient cooling performance between non-selective, mono-selective, and dual-selective thermal emitters in ideal scenarios ($h = 0 \text{ W m}^{-2} \text{ K}^{-1}$).

1) Dual-selective vs. Mono-selective

As the atmospheric transparency of the second atmospheric window is slightly less than 1 compared with mono-selective emitters, there is also an additional (albeit small) atmospheric heating effect for dual-selective emitters that cannot be ignored under ideal conditions. Therefore, in ideal environments, mono-selective thermal emitters have a slightly higher temperature reduction than dual-selective emitters.

2) Dual-selective vs. Non-selective

As for non-selective thermal emitters, even if they dissipate heat quickly (with high cooling power), the trade-off is that most of their heat in the non-window wavebands is absorbed and blocked by the atmosphere. **As a result, in ideal scenarios ($h = 0 \text{ W m}^{-2} \text{ K}^{-1}$), non-selective emitters emit the least total amount of heat to the outer space compared to the other two types of thermal emitters, resulting in the smallest temperature reduction ultimately achieved.** In contrast, selective thermal emitters (both for mono- and dual-selective) can dissipate most of their heat into the outer space, although they cool slowly, but can achieve a much higher temperature reduction once steady state is reached. Therefore, in ideal environments, dual-selective emitters have a much higher temperature reduction than non-selective emitters.

2. The detailed explanations for that the dual-selective RC system surpasses both non-selective and mono-selective counterparts in real scenarios.

In real scenarios ($h \geq 1 \text{ W m}^{-2} \text{ K}^{-1}$), the presence of non-radiative heat (e.g., convection) will induce heat exchange between the radiative cooling system and the surroundings, counteracting the radiative cooling effect (i.e., heat leakage) and drastically weakening the subambient cooling performance of thermal emitters (Fig. 1c, e and Supplementary Fig. 2). **Importantly, the higher the net cooling power (the faster they dissipate heat), the less the subambient cooling performance of a thermal emitter is affected by non-radiative heat effects.**

Comparing mono-selective thermal emitters with non-selective thermal emitters, when the non-radiative effect is small ($h = 1\text{--}2 \text{ W m}^{-2} \text{ K}^{-1}$), the subambient cooling capacity of the former is greater than that of the latter due to the fact that the former can exclude atmospheric parasitic heat and is subject to less attenuation by non-radiative heat. If the non-radiative effect is significant ($h \geq 3 \text{ W m}^{-2} \text{ K}^{-1}$), the attenuation of the subambient cooling capacity of the former is much greater than that of the latter due to its much lower net cooling power than that of the latter, resulting in the subambient cooling performance of the former being less than that of the latter.

With regard to dual- and mono-selective thermal emitters, although both are capable of eliminating atmospheric parasitic heat, due to the much higher net cooling power of the former, the attenuation of the subambient cooling capacity of the former affected by non-radiative heat is much less than that of the latter. Therefore, in real environments ($h \geq 1 \text{ W m}^{-2} \text{ K}^{-1}$), dual-selective thermal emitters have a much better subambient cooling performance than mono-selective thermal emitters.

In addition, when comparing the dual-selective with non-selective thermal emitters, although they have relatively similar net cooling power (i.e., relatively similar heat dissipation rate), the former can exclude atmospheric parasitic heat. Therefore, dual-selective thermal emitters have a better subambient cooling performance than non-selective thermal emitters in real environments ($h = 1\text{--}6 \text{ W m}^{-2} \text{ K}^{-1}$).

In summary, **dual-selective thermal emitters can exclude atmospheric parasitic heat and exhibit high cooling power, combining the advantages of non-selective and mono-selective thermal emitters and avoiding their disadvantages** (inability to exclude atmospheric parasitic heat for non-selective thermal emitters and low net cooling power for mono-selective thermal emitters). As a result, in real arid environments, the subambient radiative cooling performance of dual-selective

thermal emitters is notably better than that of non-selective and mono-selective counterparts.

The corresponding revisions have been added to the revised Manuscript (Pages 5,6) and revised Supplementary Materials (Supplementary texts 2,3, Pages 6-9), which are shown as follows.

“However, in real scenarios ($h \geq 1 \text{ W m}^{-2} \text{ K}^{-1}$)^{35,36}, the subambient cooling performance of the dual-selective RC model is significantly better than that of the non-selective and mono-selective RC models. For example, at $h = 1\text{--}6 \text{ W m}^{-2} \text{ K}^{-1}$, the dual-selective RC model was 1.0–10.9 °C cooler than the non-selective model, and 4.2–5.5 °C cooler than the mono-selective model (Fig. 1e, Supplementary text 3 and Supplementary Fig. 3, and Supplementary Table 1).” (Pages 5,6 of the revised Manuscript)

“Supplementary text 2. Relationship between the net cooling power and the subambient temperature reduction for dual-selective, mono-selective, and mono-selective thermal emitters.

..... Obviously, high cooling power does not imply large temperature reduction, and our ultimate goal in developing radiative cooling technology is to achieve high temperature reduction in most practical applications. Therefore, we prefer to use the subambient temperature reduction rather than the net cooling power to evaluate the radiative cooling performance of thermal emitters.....

As analyzed above, while non-selective thermal emitters dissipate heat quickly (with high cooling power), the trade-off is that most of the heat in the non-window band is absorbed/blocked by the atmosphere and returns to re-heat the emitters (i.e., atmospheric parasitic heat). As a result, in ideal environments ($h = 0 \text{ W m}^{-2} \text{ K}^{-1}$), non-selective emitters emit the least total amount of heat to the outer space compared to the other two types of thermal emitters, resulting in the smallest temperature reduction ultimately achieved.

In contrast, selective thermal emitters can dissipate most of their heat into the outer space, although they cool down slowly, but can achieve a much higher temperature reduction once the steady state is reached. In addition, since the atmospheric transparency of the second atmospheric window is slightly less than 1, comparing with

mono-selective emitters, there is also an additional (albeit small) atmospheric heating effect for dual-selective emitters that cannot be ignored under ideal conditions. Therefore, in ideal environments, mono-selective thermal emitters have a slightly higher temperature reduction than dual-selective emitters (Fig. 1e).” (Pages 6,7 of the revised Supplementary Materials)

“Supplementary text 3. Effect of non-radiative heat on the subambient cooling performance of dual-selective, mono-selective, and mono-selective thermal emitters.

In real scenarios ($h \geq 1 \text{ W m}^{-2} \text{ K}^{-1}$), the presence of non-radiative heat (e.g., convection) will induce heat exchange between the radiative cooling system and the surroundings, counteracting the radiative cooling effect (i.e., heat leakage) and drastically weakening the subambient cooling performance of thermal emitters (Fig. 1c,e and Supplementary Fig. 2). Importantly, the higher the net cooling power (the faster they dissipate heat), the less the subambient cooling performance of a thermal emitter is affected by non-radiative heat effects.

Comparing mono-selective thermal emitters with non-selective thermal emitters, when the non-radiative effect is small ($h = 1\text{--}2 \text{ W m}^{-2} \text{ K}^{-1}$), the subambient cooling capacity of the former is greater than that of the latter due to the fact that the former can exclude atmospheric parasitic heat and is subject to less attenuation by non-radiative heat. If the non-radiative effect is significant ($h \geq 3 \text{ W m}^{-2} \text{ K}^{-1}$), the attenuation of the subambient cooling capacity of the former is much greater than that of the latter due to its much lower net cooling power than that of the latter, resulting in the subambient cooling performance of the former being less than that of the latter.

With regard to dual- and mono-selective thermal emitters, although both are capable of eliminating atmospheric parasitic heat, due to the much higher net cooling power of the former, the attenuation of the subambient cooling capacity of the former affected by non-radiative heat is much less than that of the latter. Therefore, in real environments ($h \geq 1 \text{ W m}^{-2} \text{ K}^{-1}$), dual-selective thermal emitters have a much better subambient cooling performance than mono-selective thermal emitters.

In addition, when comparing the dual-selective with non-selective thermal emitters, although they have relatively similar net cooling power (i.e., relatively similar heat

dissipation rate), the former can exclude atmospheric parasitic heat. Therefore, dual-selective thermal emitters have a better subambient cooling performance than non-selective thermal emitters in real environments ($h = 1-6 \text{ W m}^{-2} \text{ K}^{-1}$).

In summary, dual-selective thermal emitters can exclude atmospheric parasitic heat and exhibit high cooling power, combining the advantages of non-selective and mono-selective thermal emitters and avoiding their disadvantages (inability to exclude atmospheric parasitic heat for non-selective thermal emitters and low net cooling power for mono-selective thermal emitters). As a result, in real arid environments, the subambient radiative cooling performance of dual-selective thermal emitters is notably better than that of non-selective and mono-selective counterparts.” (Pages 8,9 of the revised Supplementary Materials)

Comment 4. In Figure 2e, the author notes that they have determined fiber and particle sizes based on scattering efficiency simulations. To bolster this claim, the author should consider including a comparison with the reflection data of a neat Al foil in Figure 2f.

Reply: Thank you for your suggestion. The reflectance spectrum of the neat Al foil in the solar waveband (0.3–2.5 μm) has been used in the revised version for comparison with that of the POM-PTFE-Al sample to support the claim that the fiber and particle sizes were determined by the Mie scattering simulations. We have also added the detailed explanations to further clear our design principles for the fiber- and particle-size distributions. The details are as follows.

1. Comparison of the solar reflectance spectra of the POM-PTFE-Al sample and the pure Al foil

As shown in Fig. R2-5, the POM-PTFE-Al sample showed a much higher solar reflectance (95.4%) than that of the pure Al foil (88.7%), indicating that the POM-PTFE electrospun film was indeed effective in enhancing the solar reflectance of the dual-selective sample. The further analyzes show that its higher solar reflectance was due to its higher reflectance in both the ultraviolet-visible (UV-Vis, 95.5%) and the near-infrared (NIR, 95.2%) wavebands than the pure Al foil (85.5% and 92.5%, respectively). The much higher UV-Vis reflectance (0.3–0.76 μm) of POM-PTFE-Al than the pure Al foil was mainly due to the nano-design of the POM nanofibers and the higher NIR reflectance (0.76–2.5 μm) was mainly due to the micro-design of the PTFE microparticles. These results demonstrate that the enhancement of the solar reflectance by the nano/micron structure (including POM nanofibers and PTFE microparticles) determined from

Mie scattering simulations is remarkable.

It should be noted that the pure Al foil, as a good heat reflector, also showed a high NIR reflectance in the 0.76–2.5 μm waveband (92.5%), indicating that the high NIR reflectance of the dual-selective sample is due to a combined effect of the POM-PTFE design (mainly in the 0.76–1.2 μm waveband) and the Al foil (mainly in the 1.2–2.5 μm waveband).

To avoid possible confusion and misinterpretation when the data of the Al foil are added directly to the Fig. 2f, the figure containing the solar reflectance data for both the neat Al foil and the POM-PTFE-Al sample has been added to the Supplementary Materials (Supplementary Fig. 19 in the revised Supplementary Materials).

Fig. R2-5. Comparison of the solar reflectance spectra of the POM-PTFE-Al sample and the pure Al foil in the 0.3–2.5 μm waveband. a, Solar reflectance spectra of both the POM-PTFE-Al and the Al foil. **b,** Corresponding average reflectance of total (0.3–2.5 μm), UV-vis (0.3–0.76 μm), and NIR (0.76–2.5 μm) wavebands, respectively. (Supplementary Fig. 19 in the revised Supplementary Materials)

2. Design principles for the sizes of PTFE particles and POM fibers

According to the Mie scattering theory, particle size is indeed the key to achieving high solar reflectance for a particle-based daytime radiative cooling material. We have chosen the appropriate particle size distribution of PTFE particles according to the following principles.

1) First, according to our theoretical calculations for PTFE particles based on Mie theory (**Fig. 2b** in the revised manuscript), for a given thickness of a radiative cooler which is fixed, the highest solar reflectance can be achieved in an air medium when the size distribution of PTFE particles is around 0.2–3.0 μm (close to the solar waveband). It should be noted that the result was obtained

for PTFE particles in an air medium where there is a large difference in refractive indices between PTFE and air, resulting in strong scattering at the polymer/air interface. However, for the POM-PTFE complexes-based samples, considering the matching of POM fibers and PTFE particles in the samples, we prefer to select PTFE particles with a size larger than that of POM fibers. The reason is that as-used polymers, *i.e.*, PTFE and POM, both have similar refractive indices (~ 1.5 , *Astrophys Space Sci* 39, L13-L18, (1976); *Mon. Not. R. Astron. Soc.* 175, 197-207, (1976); *Optics InfoBase Conference Papers*, (2007)). If the size of the PTFE particle is similar or smaller than that of the POM fiber (*e.g.*, using nano-sized PTFE particles to prepare the POM-PTFE electrospun film), the Mie scattering effect of the particles will be masked by the POM fibers because the PTFE nanoparticles are embedded in the POM fibers (*i.e.*, the PTFE particles are not in the air medium). Therefore, the PTFE particles with a larger size distribution than that of the POM fibers were used to prepare the dual-selective samples.

2) In addition, our previous work has shown that nano-sized POM fibers with diameter distribution close to the main waveband of sunlight (0.2–1.0 μm) can be easily produced by electrospinning (*Nat. Sustain.*, 2023, <https://doi.org/10.1038/s41893-023-01200-x>).

3) Finally, nanoparticle-based products are susceptible to health risks due to the inevitable inhalation of the human body during production and use. Besides, although both nano- and micron-sized PTFE particles are commercially available, nano-sized PTFE particles are much more expensive than micron-sized PTFE particles.

Therefore, taking the Mie scattering effect, compatibility with POM fibers, safety, and cost of PTFE particles into account, the POM fibers with distribution of 0.2–1.0 μm and PTFE particles with distribution of 1.0–3.0 μm were used to prepare the dual-selective samples.

The corresponding revisions have been added to the main text of the revised Manuscript (Pages 10,11) and revised Supplementary Materials (Supplementary text 6 and Supplementary Fig. 19) as follows.

“As shown in Fig. 2b, a strong scattering (high reflectance) in the solar waveband can be achieved when the fiber or particle diameter distribution is close to the solar waveband (200–3000 nm), providing a clear guide for hierarchically structural designs (Supplementary text 6). Therefore, a hierarchical POM-PTFE covered on a metal substrate is expected to be an ideal dual-selective thermal emitter with a high solar

reflectance.” (Page 10 of the revised manuscript)

“The appropriate size distribution close to the wavelength range of the solar spectrum, the disordered arrangement of the fibers, and the highly reflective Al foil render the POM-PTFE-Al thermal emitter with a high solar reflectance of up to 95.4% in the 0.3–2.5 μm waveband (Fig. 2f and Supplementary Fig. 19).” (Page 11 of the revised manuscript)

Supplementary Fig. 19. Comparison of the solar reflectance spectra of the POM-PTFE-Al sample and the pure Al foil in the 0.3–2.5 μm waveband. *a*, Solar reflectance spectra of both the POM-PTFE-Al and the Al foil. *b*, Corresponding average reflectance of total (0.3–2.5 μm), UV-vis (0.3–0.76 μm), and NIR (0.76–2.5 μm) wavebands, respectively. As can be seen, the POM-PTFE-Al sample showed a much higher solar reflectance (95.4%) than that of the pure Al foil (88.7%), indicating that the POM-PTFE electrospun film was indeed effective in enhancing the solar reflectance of the dual-selective sample. The further analyzes show that its higher solar reflectance was due to its higher reflectance in both the ultraviolet-visible (UV-Vis, 95.5%) and the near-infrared (NIR, 95.2%) wavebands than the pure Al foil (85.5% and 92.5%, respectively). The much higher UV-Vis reflectance (0.3–0.76 μm) of POM-PTFE-Al than the pure Al foil was mainly due to the nano-design of the POM nanofibers and the higher NIR reflectance (0.76–2.5 μm) was mainly due to the micro-design of the PTFE microparticles. These results demonstrate that the enhancement of the solar reflectance by the nano/micron structure (including POM nanofibers and PTFE microparticles) determined from Mie scattering simulations is remarkable.” (Page 41 of the revised Supplementary Materials)

“Supplementary text 6. Design principles for the diameter distribution of PTFE particles and POM fibers

According to the Mie scattering theory, the diameter distribution of the particles or fibers is the key to achieving a high solar reflectance of a particle- or fiber- based thermal emitter (for daytime radiative cooling). We have selected the appropriate size distribution of PTFE particles and POM fibers according to the following principles.

1) First, according to our theoretical calculations for PTFE particles based on the Mie scattering theory (Fig. 2b), for a given thickness of a radiative cooler, the highest solar reflectance can be achieved in an air medium when the size distribution of PTFE particles is around 0.2–3.0 μm (which is close to the solar waveband). It should be noted that the result was obtained for PTFE particles in an air medium where there is a large difference in refractive indices between PTFE and air, resulting in strong scattering at the polymer/air interface. However, for the POM-PTFE complexes-based samples, considering the matching of POM fibers and PTFE particles in the samples, we prefer to select PTFE particles with a size larger than that of POM fibers. The reason is that the as-used polymers, i.e., PTFE and POM, both have similar refractive indices (~ 1.5)^{9,11,12}. If the size of the PTFE particle is similar or smaller than that of the POM fiber (e.g., using nano-sized PTFE particles to prepare the POM-PTFE electrospun film), the Mie scattering effect of the particles will be masked by the POM fibers because the PTFE nanoparticles are embedded in the POM fibers (i.e., the PTFE particles are not in the air medium). Therefore, the PTFE particles with a larger size distribution than that of the POM fibers were used to prepare the dual-selective samples.

2) In addition, our previous work has shown that nano-sized POM fibers with diameter distribution close to the main waveband of sunlight (0.2–1.0 μm) can be easily produced by electrospinning¹³.

3) Finally, nanoparticle-based products are susceptible to health risks due to the inevitable inhalation of the human body during production and use. Besides, although both nano- and micron-sized PTFE particles are commercially available, nano-sized PTFE particles are much more expensive than micron-sized PTFE particles.

Therefore, taking the Mie scattering effect, compatibility with POM fibers, safety, and cost of PTFE particles into account, the POM fibers with distribution of 0.2–1.0 μm and PTFE particles with distribution of 1.0–3.0 μm were used to prepare the dual-selective

samples.” (Pages 13,14 of the revised Supplementary Materials)

Comment 5. The author should detail the quantitative method employed for characterizing fiber and particle sizes. This should encompass information regarding the number of samples characterized and averaged (not relative), as well as the specific software or methodologies employed in the analysis.

Reply: Thank you for your suggestion. We have added a quantitative test method for fiber and particle size in the revised version as follows.

The software used for quantitative size statistics was ImageJ, based on the SEM images of the POM-PTFE electrospun film (**Fig. R2-6**, Supplementary Fig. 15 in the revised Supplementary Materials). The number of samples of both fibers and particles was 100 (randomly selected from the SEM images). The ratio of the number of particles/fibers to the total number of samples in each size range was counted (*i.e.*, relative count, **Fig. R2-7**, Supplementary Fig. 18 in the revised Supplementary Materials). This allows us to directly obtain the distribution of fibers and particles, which is a commonly used representation (*Nat. Nanotechnol.* 16, 153-158, (2021)). The final average diameter values obtained for POM fibers and PTFE particles are the average sizes of the 100 samples measured with ImageJ, which are 502 ± 154 nm and 1.8 ± 0.49 μ m, respectively.

Fig. R2-6. SEM images of a dual-selective POM-PTFE film, which has a bead-like fiber structure consisting of POM nanofibers and PTFE micron particles. (**Supplementary Fig. 15** in the revised Supplementary Materials)

Fig. R2-7. Statistical distribution of the diameters of the POM nanofibers and PTFE particles inside the dual-selective POM-PTFE film. (Supplementary Fig. 18 in the revised Supplementary Materials)

The corresponding revisions have been added to the figure caption of Supplementary Fig. 18 in the Supplementary Materials as follow.

“Supplementary Fig. 18. Statistical distribution of the diameters of the POM nanofibers and PTFE particles inside the dual-selective POM-PTFE film. ImageJ software was used to perform these statistics based on the SEM images of POM-PTFE electrospun film (Supplementary Fig. 15). The number of samples of both fibers and particles was 100 (randomly selected from the SEM images), and the ratio of the number of particles/fibers to the total number of samples in each size range was counted (i.e., relative count), so that the size distribution of fibers and particles could be obtained directly. The final average diameter values obtained for POM fibers and PTFE particles are the average sizes of the 100 samples measured with ImageJ, which are 502 ± 154 nm and 1.8 ± 0.49 μm, respectively.” (Page 38 of the Supplementary Materials)

Comment 6. In Figure 4b, the author presents the cooling power of both dual-selective and mono-selective RC systems based on real experiments. To ensure a fair comparison and maintain consistency throughout the manuscript, it would be advisable to include the cooling power data for the non-selective RC system as well. Additionally, it would be valuable if the author could explore potential correlations between the simulated cooling powers and the measured cooling powers, providing further insights into the accuracy and reliability of the simulations in predicting real-world

performance.

Reply: Thank you for your valuable suggestion. We have added the cooling power data of the non-selective sample in the revised version as a proof of concept, and also analyzed the potential correlations between the simulated cooling powers and the measured cooling powers. The details are as follows.

1. Experimental comparison of the net cooling power between the non-selective sample and the dual-selective sample.

As shown in the simulations in the manuscript, dual-selective thermal emitters exhibit higher subambient cooling performance than mono- and non-selective counterparts in real arid environments. However, the reasons why their cooling performance is higher than that of the other two types of thermal emitters are different. Specifically, the higher performance than mono-selective thermal emitters is due to that dual-selective emitters have a much higher net cooling power than mono-selective thermal emitters, which has been demonstrated by the cooling power measurements (Fig. 4b in the revised manuscript). **In contrast, the better performance of dual-selective thermal emitters compared with non-selective thermal emitters is because of the ability of dual-selective emitters to exclude atmospheric parasitic heat, not because of their higher net cooling power (which is in fact slightly lower than that of non-selective thermal emitters). This is also the reason why the cooling power data of the non-selective sample is not included in the raw Fig. 4b of the manuscript.**

To demonstrate this, we also compared experimentally the net cooling power of the non-selective and dual-selective samples in a similar arid environment ($RH = \sim 12\%$, **Figs. R2-8 and R2-9**). **The results show that under strong solar irradiation ($\sim 700 \text{ W m}^{-2}$, Figs. R2-9a), the dual-selective POM-PTFE-Al exhibited slightly lower net cooling power ($131.2 \pm 19.7 \text{ W m}^{-2}$), than the non-selective PVDF ($154.5 \pm 11.0 \text{ W m}^{-2}$), in agreement with our simulations.**

It is worth noting that the cooling power of the dual-selective sample here is lower than the result in Fig. 4b ($151.8 \pm 13.1 \text{ W m}^{-2}$, Fig. 4b), which is due to the two tests with different environmental conditions between the two cooling power measurements, including ambient temperature, humidity, wind speed, and solar irradiation (mainly lower temperatures and higher wind speeds, resulting in lower net cooling power).

Fig. R2-8. Multi-scenario human body cooling measurements. (Supplementary Fig. 35a in the revised Supplementary Materials)

Fig. R2-9. Multi-scenario human body cooling measurements. (Supplementary Fig. 34 in the revised Supplementary Materials)

2. Correlation between the simulated and the measured cooling powers

To explore the potential correlations between the simulated and measured results for the net cooling powers, we have simulated the net cooling powers of these samples according to the ambient temperature of the real measurement conditions and compared them with the real experimental results.

As shown in **Fig. R2-10**, the measurements for the dual-selective and mono-selective samples

($T_{\text{ambient}} = \sim 38 \text{ }^{\circ}\text{C}$, 7 September 2022) showed that the dual-selective POM-PTFE-Al exhibited an ultra-high net cooling power ($151.8 \pm 13.1 \text{ W m}^{-2}$), which was much larger than that of the mono-selective POM ($87.9 \pm 9.4 \text{ W m}^{-2}$). Their corresponding theoretical cooling powers (excluding solar heat and $T_{\text{ambient}} = 38 \text{ }^{\circ}\text{C}$) were 208.8 W m^{-2} and 151.2 W m^{-2} , respectively. **Taking the cooling power compensated by solar heat into consideration (840 W m^{-2} for solar irradiation, 38.6 W m^{-2} for the dual-selective sample and 47.9 W m^{-2} for the mono-selective sample), the actual measured net cooling powers of the dual-selective and mono-selective samples were $190.4 \pm 13.1 \text{ W m}^{-2}$ and $135.8 \pm 9.4 \text{ W m}^{-2}$, respectively, which were close to the theoretical values.**

As shown in Fig. R2-11, the measurements for the dual-selective POM-PTFE-Al and non-selective PVDF in an environment with lower ambient temperature ($T_{\text{ambient}} = \sim 31 \text{ }^{\circ}\text{C}$, 9 September 2022) showed that the dual-selective sample exhibited a high net cooling power ($131.2 \pm 19.7 \text{ W m}^{-2}$), which was close to that of the non-selective samples ($154.5 \pm 11.0 \text{ W m}^{-2}$). Their corresponding theoretical cooling powers (excluding solar heat and $T_{\text{ambient}} = 31 \text{ }^{\circ}\text{C}$) were 190.4 W m^{-2} and 221.0 W m^{-2} , respectively. **Taking into account the cooling power compensated by solar heat (700 W m^{-2} for solar irradiation, 32.2 W m^{-2} for the dual-selective sample and 34.3 W m^{-2} for the non-selective sample), the actual measured net cooling powers of the dual-selective and mono-selective samples were $163.4 \pm 19.7 \text{ W m}^{-2}$ and $188.8 \pm 11.0 \text{ W m}^{-2}$, respectively, which were also close to their theoretical values.**

Fig. R2-10. Average cooling power of the dual-selective POM-PTFE-Al and mono-selective POM-Al and the corresponding theoretical cooling power. (Supplementary Fig. 33b in the revised Supplementary Materials)

Fig. R2-11. Average cooling power of the dual-selective POM-PTFE-Al and non-selective PVDF and the corresponding theoretical cooling power. (Supplementary Fig. 35b in the revised Supplementary Materials)

In conclusion, our experimental results are lower than the theoretical values, mainly due to the absorption of solar heat. When the solar heating effect is taken into account, they are close to the theoretical values. Nevertheless, there are still deviations between the theoretical and experimental values due to unavoidable deviations, such as humidity, wind and instrumentation conditions in the actual tests.

The corresponding revisions have been added to the main text of the revised Manuscript (Page 15) and revised Supplementary Materials (Supplementary texts 11,12, Supplementary Figs. 34,35), which are shown as follows.

“In comparison, the dual-selective sample was shown to have a cooling power close to the non-selective one (Supplementary text 11 and Supplementary Figs. 34,35). These results indicate that the as-designed dual-selective thermal emitter has an enhanced daytime radiative cooling performance over the existing mono-selective and non-selective thermal emitters in real scenarios, especially in arid climates, which is in good agreement with the theoretical analysis (Supplementary text 12).” (Page 15 of the revised Manuscript)

“Supplementary text 11. Experimental comparison of the net cooling power between the non-selective sample and the dual-selective sample.

As shown in the simulations in this work (Fig.1), dual-selective thermal emitters

exhibit higher subambient cooling performance than mono- and non-selective counterparts in real arid environments. However, the reasons why their subambient cooling performance is higher than that of the other two types of thermal emitters are different. Specifically, the higher performance than mono-selective thermal emitters is due to that dual-selective emitters have a much higher net cooling power than mono-selective thermal emitters, which has been demonstrated by the cooling power measurements (Fig. 4b in the revised manuscript). In contrast, the higher performance of dual-selective thermal emitters compared to non-selective thermal emitters is due to the ability of dual-selective emitters to exclude atmospheric parasitic heat, not to their higher net cooling power (which is in fact slightly lower than that of non-selective thermal emitters). This is also the reason why the cooling power data of the non-selective sample is not included in the raw Fig. 4b. To demonstrate this, we also compared experimentally the net cooling power of the non-selective and dual-selective samples in a similar arid environment (RH = ~12%, Supplementary Figs. 34 and 35). The results show that under strong solar irradiation (~700 W m⁻²), the dual-selective POM-PTFE-Al exhibited slightly lower net cooling power (131.2±19.7 W m⁻²), than the non-selective PVDF (154.5±11.0 W m⁻²), in agreement with our simulations.

It is worth noting that the cooling power of the dual-selective sample here is lower than the result in Fig. 4b (151.8 ± 13.1 W m⁻², Fig. 4b), which is due to the two tests with different environmental conditions between the two cooling power measurements, including ambient temperature, humidity, wind speed, and solar irradiation (mainly lower temperatures and higher wind speeds, resulting in lower net cooling power).” (Page 19 of the revised Supplementary Materials)

Supplementary text 12. Correlation between the simulated and the measured cooling powers.

To explore the potential correlations between the simulated and measured results for the net cooling powers, we have simulated the net cooling powers of these samples according to the ambient temperature of the real measurement conditions and compared them with the real experimental results.

As shown in Supplementary Fig. 33b, the measurements for the dual-selective and

mono-selective samples ($T_{\text{ambient}} = \sim 38 \text{ }^{\circ}\text{C}$, 7 September 2022) showed that the dual-selective POM-PTFE-Al exhibited an ultra-high net cooling power ($151.8 \pm 13.1 \text{ W m}^{-2}$), which was much larger than that of the mono-selective POM ($87.9 \pm 9.4 \text{ W m}^{-2}$). Their corresponding theoretical cooling powers (excluding solar heat and $T_{\text{ambient}} = 38 \text{ }^{\circ}\text{C}$) were 208.8 W m^{-2} and 151.2 W m^{-2} , respectively. Taking the cooling power compensated by solar heat into consideration (840 W m^{-2} for solar irradiation, 38.6 W m^{-2} for the dual-selective sample and 47.9 W m^{-2} for the mono-selective sample), the actual measured net cooling powers of the dual-selective and mono-selective samples were $190.4 \pm 13.1 \text{ W m}^{-2}$ and $135.8 \pm 9.4 \text{ W m}^{-2}$, respectively, which were close to the theoretical values.

As shown in Supplementary Fig. 35b, the measurements for the dual-selective POM-PTFE-Al and non-selective PVDF in an environment with lower ambient temperature ($T_{\text{ambient}} = \sim 31 \text{ }^{\circ}\text{C}$, 9 September 2022) showed that the dual-selective sample exhibited a high net cooling power ($131.2 \pm 19.7 \text{ W m}^{-2}$), which was close to that of the non-selective samples ($154.5 \pm 11.0 \text{ W m}^{-2}$). Their corresponding theoretical cooling powers (excluding solar heat and $T_{\text{ambient}} = 31 \text{ }^{\circ}\text{C}$) were 190.4 W m^{-2} and 221.0 W m^{-2} , respectively. Taking into account the cooling power compensated by solar heat (700 W m^{-2} for solar irradiation, 32.2 W m^{-2} for the dual-selective sample and 34.3 W m^{-2} for the non-selective sample), the actual measured net cooling powers of the dual-selective and mono-selective samples were $163.4 \pm 19.7 \text{ W m}^{-2}$ and $188.8 \pm 11.0 \text{ W m}^{-2}$, respectively, which were also close to their theoretical values.

In conclusion, our experimental results are lower than the theoretical values, mainly due to the absorption of solar heat. When the solar heating effect is taken into account, they are close to the theoretical values. Nevertheless, there are still deviations between the theoretical and experimental values due to unavoidable deviations, such as humidity, wind and instrumentation conditions in the actual tests. (Page 20 of the revised Supplementary Materials)

Supplementary Fig. 34. Ambient conditions of the cooling power measurements for the dual-selective and non-selective films (in the Ulan Buh Desert, 9 September 2022). a, Solar irradiation. b, Ambient air temperature and relative humidity. c, Wind speed.” (Page 55 of the revised Supplementary Materials)

Supplementary Fig. 35. Cooling power measurements (in the Ulan Buh Desert, $\text{RH} \approx 12\%$, 9 September 2022). a, Real-time cooling power measurements of the dual-selective POM-PTFE-Al and non-selective PVDF from 02:00 a.m. to 02:50 p.m. b, Correlation between the simulated and the measured cooling powers, including the average cooling power of the dual-selective POM-PTFE-Al and non-selective PVDF and the corresponding theoretical cooling power. Error bars in b indicate measurement variations of the samples at different times and show the mean \pm s.d. ($n = 81$).” (Page 56 of the revised Supplementary Materials)

Comment 7. The author highlights that the dual-selective radiative cooling (RC) system exhibits better color compatibility. However, it's worth noting that for the demonstrated colored RC, the cooling performance does not appear to be superior. In particular, the blue-colored RC seems to

exhibit heating rather than cooling, likely due to its high absorbance in the 0.6-0.8 μm range. It might not be entirely appropriate to claim better color compatibility if the demonstrated RC systems are not achieving actual cooling, especially considering the existence of research papers showcasing effective cooling performance with colored materials. To strengthen their argument, the author should provide additional data or experimental results that demonstrate the actual cooling behavior of the blue-colored RC.

Reply: Thank you for your question. Based on your suggestion, we have added additional experiments (for the blue dual-selective sample) and reviewed the latest research progress in color radiative cooling, as well as carefully comparing the cooling performance between the previous work and this work to demonstrate the high color compatibility of the dual-selective thermal emitter. The details are as follows.

1. Subambient radiative cooling vs. above-ambient radiative cooling.

Firstly, to avoid any misunderstanding, we would like to clarify that radiative cooling can be divided into **subambient radiative cooling** (cooling to below the ambient temperature) and **above-ambient radiative cooling** (cooling only to above the ambient temperature, *Nat. Nanotechnol.* 16, 1342–1348 (2021), *Adv. Funct. Mater.* 32, (2022)). The latter has an input heating power (such as solar heat or human body heat) higher than the radiative cooling power, and actually exhibits a "heating mode" (as you point out with the blue-colored RC sample). Therefore, it is unable to achieve subambient cooling. Nevertheless, these materials are still useful as cooling materials due to their radiative cooling effect, **which is remarkable compared with the existing commercial or non-photonically designed counterparts**. In this work, **the blue dual-selective sample belongs to an above-ambient radiative cooling material**.

2. Measurement of the radiative cooling performance of the blue dual-selective thermal emitter.

To demonstrate the radiative cooling effect of the colored dual-selective thermal emitters, in addition to the yellow dual-selective sample, the cooling performance of the blue dual-selective sample was tested and compared with the commercial counterpart (blue PE covered commercial white paint). The IR camera tests in a low-humidity environment (RH \sim 18% and solar \sim 500 W m^{-2} , Figs. R2-12 and R2-13) showed that, under strong sunlight, the blue PE covered dual-selective POM-PTFE-Al thermal emitter exhibited significantly lower cooling performance (5-8 $^{\circ}\text{C}$ cooler) than the blue PE covered commercial white paint (Fig. R2-12b and Supplementary Movie 5). **These results**

showed that although subambient cooling could not be achieved, the blue sample still had significantly better cooling performance than the commercial counterpart due to the above-ambient radiative cooling effect.

Fig. R2-12. Actual outdoor cooling performance test for the blue dual-selective thermal emitter covered asphalt compared to the blue PE film covered white painted asphalt, including their photograph (a) and IR images (b).

Fig. R2-13. Ambient conditions of the outdoor IR measurement for a blue dual-selective sample.

a, Solar irradiation. **b**, Relative humidity. **c**, Wind speed.

3. Comparison the cooling performance between the previous work and this work

To date, most existing colored radiation cooling materials are unable to achieve subambient cooling (*i.e.*, only above-ambient cooling) due to the inevitable absorption of visible light by colored materials (*Sol. RRL* 2023, 2300512; *Nat. Commun.* 9, 4240, (2018); *Sci Adv* 6, eaaz5413, (2020)). For example, Yang group (*Sci Adv* 6, eaaz5413, (2020)) developed a colored radiative cooler which comprised a visible-absorptive colored layer atop and a solar-scattering underlayer (**Figs. R2-14a, b**). The radiative cooler attained higher near-to-short wavelength infrared reflectance (**Fig. R2-14c**) than the commercial paints of the same color and stayed cooler by as much as 3.0 to 15.6 °C under strong sunlight (**Fig. R2-14d**). However, these colored RC materials cannot achieve subambient cooling (**Figs. R2-14d**) due to the massive solar thermal load (low solar reflectance mainly originated from visible absorption).

Recently, some excellent work has indeed been done to achieve subambient colored radiative cooling (*Sci Bull* 67, 1874-1881, (2022), *Nano Lett.* 2022, 22, 12, 4925–4932; *Adv. Sci.* 2022, 9, 2202061). For example, Zhu and co-workers (*Sci. Bull.* 67, 1874-1881, (2022)) developed a perovskite quantum dot-based radiative cooler that exploited the photoluminescence effect of quantum dots in the visible waveband to achieve both vibrant colors and high solar reflectance, thereby enabling 5.4–2.2 °C subambient cooling of colored RC materials (**Figs. R2-14e-h**). However, the perovskite quantum dot is high cost and harmful to human body, thus cannot be scale-up. Ma and co-workers (*Nano Lett.* 2022, 22, 12, 4925–4932) developed a structural color based flexible colored radiative cooler, and achieved subambient cooling of 4 °C under strong sunlight (**Figs. R2-14i-k**). However, complex photonic crystal and demanding preparation processes make it difficult to achieve large-area preparation and large-scale application of colored radiative cooling materials.

Fig. R2-14. Colored radiative cooling achieved by (a-d) Yang group (*Sci Adv* 6, eaaz5413, (2020)), (e-h) Zhu group (*Sci. Bull.* 67, 1874-1881, (2022)), and (i-l) Ma group (*Nano Lett.* 2022, 22, 12, 4925–4932).

In contrast, this work demonstrates that efficient and scalable colored radiative cooling can be achieved by simply covering on the surface of the dual-selective thermal emitter with a commercially available colored PE film. As a result, **the red and yellow radiative coolers achieve subambient radiative cooling of 5° and 3.3 °C, respectively (which is close to what has been reported in the literature).** The blue radiative cooler, although not shown as being able to achieve subambient radiative cooling in the manuscript, has a cooling performance significantly superior to that of the commercial counterpart.

In addition, as a reasonable corollary, the solar reflectance of the samples can be increased by reducing the amount of dye in the PE film (reducing the heat absorption in the visible wavelengths), which means that subambient cooling is expected to be achieved for all colors of dual-selective radiative coolers. Therefore, the dual-selective thermal emitters have a high color compatibility.

The corresponding revisions have been added to the main text (Pages 2 and 21) of the

revised Manuscript and revised Supplementary Materials (Supplementary text 13, Supplementary Figs. 52,23) as follows.

“Furthermore, even coated with different colors, the dual-selective emitter still exhibits **superior cooling performance**, indicating a high color compatibility. This work provides a scalable RC design for sustainable and energy-efficient thermal management.”

(Abstract of the revised manuscript)

“..... The same conclusion can be drawn from the IR measurement of a blue dual-selective sample (Supplementary text 13, Supplementary Figs. 52,53, and Supplementary Movie 5). These results demonstrate the high color compatibility of dual-selective POM-PTFE-Al, and therefore a high practicality in real scenarios.” (Page 21 of the revised manuscript)

“**Supplementary text 13. Measurement of the radiative cooling performance of the blue dual-selective thermal emitter.**

To demonstrate the radiative cooling performance of the colored dual-selective thermal emitters, in addition to the yellow dual-selective sample, the cooling performance of the blue dual-selective sample was tested and compared with the commercial counterpart (blue PE covered commercial white paint) in Beijing, China (8 October 2023). The IR camera tests in a low-humidity environment (RH ~18% and Solar ~500 W m⁻², Supplementary Figs. 52 and 53) showed that, under strong sunlight, the blue PE covered dual-selective POM-PTFE-Al thermal emitter exhibited significantly lower cooling performance (5–8 °C cooler) than the blue PE covered commercial white paint (Supplementary Figs. 53b and Supplementary movie 5). These results showed that although subambient cooling could not be achieved, the blue sample still had significantly better cooling performance than the commercial counterpart due to the above-ambient radiative cooling effect.” (Page 22 of the Supplementary Materials)

Supplementary Fig. 53. Ambient conditions of the outdoor IR measurement for a blue dual-selective PTFE-POM film (Beijing, China, 8 October 2023). a, Solar irradiation. b, Relative humidity. c, Wind speed.” (Page 74 of the Supplementary Materials)

Supplementary Fig. 52. Actual outdoor cooling performance test for the blue dual-selective thermal emitter covered asphalt in Beijing, China (RH = ~18%, 8 October 2023) compared to the blue PE film covered white painted asphalt, including their photograph (a) and IR images (b).” (Page 75 of the Supplementary Materials)

REVIEWERS' COMMENTS

Reviewer #1 (Remarks to the Author):

The reviews have been addressed with thorough responses to the comments received. The manuscript has been appropriately revised and amended. Therefore, I believe this manuscript meets the high standards of Nature Communications and recommend its publication.

Reviewer #2 (Remarks to the Author):

Thank you for refining the manuscript. The newly revised version has undergone meticulous editing and is now suitable for publication in "Nature Communication".